# TGFβ-activated PDHB promotes mitochondrial pyruvate metabolism and contributes to human endoderm differentiation via ATP-dependent BRG1

Liming Meng[1,6], Jing Lv[1,2,6], Ying Yi[1,6], Xianchun Lan[1], Chenchao Yan[3], Lihang Zhu[1], Jie Yang ®[1,4] ✉ & Wei Jiang ®[1,3,5] ✉

Cell fate determination is closely linked to metabolic state, yet how metabolic remodeling influences human pluripotent stem cells differentiation into three germ layers remains incompletely understood. Here, we reveal that definitive endoderm differentiation from human pluripotent stem cells requires a TGFβ-driven metabolic switch characterized by reduced lactate production and enhanced TCA cycle activity and oxidative phosphorylation, mediated by PDHB. Disruption of glucose utilization or pyruvate entry into the TCA cycle markedly impairs endoderm differentiation, whereas inhibition of lactate production enhances differentiation efficiency. Mechanistically, blockade of glucose metabolism or the TCA cycle reduces intracellular ATP levels, compromising the activity of BAF complex, an ATP-dependent chromatin remodeling complex centered on BRG1. This complex promotes chromatin accessibility and activates endodermal gene programs during differentiation. Together, these findings highlight metabolic reprogramming as a key regulator of human endoderm fate through ATP-dependent control of chromatin remodeling.

Cell fate determination is accompanied by significant metabolic changes. Glucose metabolism provides a crucial source of cellular energy and metabolic intermediates, which primarily includes two major pathways: glycolysis to produce lactate, and the pyruvate entry into mitochondria, followed by the tricarboxylic acid cycle (TCA cycle) and oxidative phosphorylation. Metabolic switches not only affect energy production but also influence the concentrations of various metabolic intermediates, such as acetyl coenzyme A (CoA) and

S-adenosylmethionine (SAM). These metabolic intermediates could further regulate the post-translational modifications of proteins, including histone, by serving as substrates of acetylation and methylation modifications, and finally affecting cell fate determination[1,2]. For instance, the mitochondrial pyruvate carrier (MPC) regulates the downstream TCA cycle during the differentiation of memory T cells, and thereby influences global H3K27ac modification levels. This modulation ultimately governs T cell differentiation and anti-tumor

[1]Department of Biological Repositories, Frontier Science Center for Immunology and Metabolism, Medical Research Institute, Zhongnan Hospital of Wuhan University, Wuhan University, Wuhan, China. [2]Hebei Key Laboratory of Bohai Rim Biomass Materials, College of Life Science, Cangzhou Normal University, Cangzhou, China. [3]The Institute of Translational Medicine and Jiangxi Province Key Laboratory of Precision Cell Therapy, The Second Affiliated Hospital, School of Basic Medical Sciences and Institute of Biomedical Innovation, Jiangxi Medical College, Nanchang University, Nanchang, China. [4]State Key Laboratory of Biocatalysis and Enzyme Engineering, School of Life Sciences, Hubei University, Wuhan, China. [5]Hubei Provincial Key Laboratory of Developmentally Originated Disease, Wuhan, China. [6]These authors contributed equally: Liming Meng, Jing Lv, Ying Yi. ✉e-mail: jyang@hubu.edu.cn; jiangw.mri@whu.edu.cn

functionality[3]. During early embryo development, the mouse two-cell embryo prefers methionine, polyamine and glutathione metabolism, whereas the blastocyst exhibits increased TCA cycle metabolites and presents a more oxidative state[4]. Interestingly, glucose is necessary for the transition from morula to blastocyst mainly via supporting anabolic processes but not energy production. Moreover, during this process, both the differentiation of the inner cell mass (ICM) into the three germ layers and the specification of the trophectoderm (TE) are highly dependent on glucose metabolism. Notably, glucose deprivation causes a severe developmental arrest and a significant loss of CDX2-positive TE cells, highlighting the essential role of glucose metabolism in early embryogenesis[5]. In addition, the high level of phospholipid unsaturation at the blastocyst stage is conserved and necessary for early mammalian embryo development. The lipid desaturases, including SCD1, are needed during the blastocyst development and implantation[6]. Overall, increasing evidence indicates a metabolic shift occurs and plays important roles during early embryo development.

Human pluripotent stem cells (PSCs), including embryonic stem cells (ESCs) derived from the ICM of human blastocysts, and induced pluripotent stem cells (iPSCs) reprogrammed from somatic cells, are defined by the abilities of self-renewal and to differentiate into three germ layers[7]. Recent studies have revealed that human PSCs exhibit a metabolic profile similar to the cancer cells, which relies predominantly on glycolysis as their primary glucose metabolism pathway even under aerobic conditions. This phenomenon is commonly referred to as aerobic glycolysis or the Warburg effect[8]. Several reports indicate that human PSCs undergo metabolic remodeling, transition from a state heavily reliant on glycolysis to a state more dependent on the TCA cycle and oxidative phosphorylation (OXPHOS), when they exit pluripotent state[9–11]. Consistent with the high aerobic glycolysis observed in PSCs, Moussaieff and colleagues found that inhibition of glycolysis can promote spontaneous differentiation and pluripotent state exit[1]. During the iPSC reprogramming, the investigators also found the upregulation of glycolysis and dynamics of mitochondrial networks, from mature somatic mitochondria with tubular and cristae-rich structures transformed into immature, spherical forms with sparse cristae in iPSCs[12].

Germ layer differentiation (i.e., ectoderm, mesoderm and endoderm), which correlates to gastrulation, is a critical event in early embryo development, thus representing a suitable model to investigate the molecular regulation during early embryo development. During lineage specification, different germ layers show different characteristics and requirements for metabolic patterns. Cliff and colleagues have discovered that a significant metabolic switch, from the high glycolysis in ESCs to a lower glycolysis and higher oxidative phosphorylation in differentiated cells, occurs during the mesoderm and definitive endoderm (DE) differentiation[13]. The following studies identified that glutamine metabolism has similar differences during germ layer differentiation. Glutamine is a major metabolic precursor of the TCA cycle, which can reflect the levels of TCA cycle and oxidative phosphorylation to a certain extent. The investigators found that exogenous glutamine is necessary for mesoderm and endoderm differentiation, whereas the ectoderm could survive in glutamine-free media[14,15]. In addition, metabolism changes also occur during extra-embryonic endoderm (XEN) cells differentiation from mouse ESCs. The XENs exhibit high glycolysis and maintain elevated lactate levels by increasing *LDHA* expression, and exogenous lactate supplement could promote the XEN differentiation[16]. Consistent with this study, another group found that the lactylation of ESRRB protein by increased lactate levels mediates the elevated differentiation efficiency[17]. Despite the metabolic switch appears during germ layer differentiation, how such a metabolic switch happens and how these precise metabolic reprogramming contributes to lineage differentiation are still largely undocumented.

Our group previously revealed the mitochondrial homeostasis plays a critical role in mesoderm and definitive endoderm differentiation from human ESCs[18,19]. Consistent with the crucial role of mitochondrial homeostasis, we recently found elevated fatty acid oxidation and reduced synthesis would result in the increased acetyl-CoA, then led to the SMAD3 acetylation to promote the expression of endoderm-related genes[20]. Inspired from previous studies, here we begin with a CRISPR-based screen targeting metabolic regulators during DE differentiation and then focus on glucose metabolism. We then characterize the glucose metabolism switch between lactate production and the TCA cycle, and demonstrate that both glucose utilization and biased pyruvate-TCA cycle flux promote DE differentiation. Disrupting the glucose metabolic switch using chemical inhibitors or genetic depletion significantly affects DE differentiation. Importantly, the inhibition of lactate production to promote pyruvate entry into the TCA cycle enhances DE differentiation efficiency. Furthermore, based on the rescue experiments using different metabolites, we figure out that the ATP production and the function of the ATP-dependent chromatin remodeling complex BAF facilitate DE differentiation.

## Results

### Glucose metabolism is essential for DE differentiation

First, we performed a CRISPR-based screening in our DE differentiation model, using a metabolism-related sgRNA library[21]. In CRISPR screening, we used CXCR4 expression as a quantitative readout to evaluate DE differentiation efficiency. As a well-established surface marker of definitive endoderm, CXCR4 (also named CD184) has been extensively utilized in previous studies to assess DE differentiation efficiency from pluripotent stem cells[20,22]. Interestingly, we obtained many genes associated with glycolysis and the TCA cycle in primary hints. And the key rate-limiting enzyme of fatty acid oxidation CPT2 also appears in these hints, which we reported recently[20], indicating the reliability of this screening (Supplementary Fig. 1a). We then performed the Gene Ontology (GO) and Kyoto Encyclopedia of Genes and Genomes (KEGG) analyses, which significantly enriched in pathways related to glucose metabolic process, glycolytic process, and electron transfer activity as well as the TCA cycle (Fig. 1a and Supplementary Fig. 1b). Moreover, the distribution of different sgRNAs targeting the individual genes shown in Supplementary Fig. 1a also confirmed our finding (Supplementary Fig. 1c). Therefore, to further investigate the impact of glucose metabolism on DE differentiation, we assessed whether the alterations in overall glucose utilization affect differentiation. We differentiated human ESCs into endoderm with different glucose concentrations (from 2.8 mM to 25 mM) and determined the differentiation efficiency. The intracellular flow cytometric analyses of key DE markers CXCR4 and SOX17 showed that higher glucose concentrations significantly resulted in higher differentiation efficiency (Fig. 1b). Moreover, RNA levels of DE-related genes *FOXA2*, *SOX17*, and *CXCR4* were upregulated along with increased glucose concentrations. In contrast, RNA expression levels of ESC-specific genes *OCT4*, *SOX2*, and *NANOG* were decreased at high glucose conditions (Fig. 1c). These results indicate that insufficient glucose supply impairs DE differentiation, and high glucose facilitates DE differentiation.

To further validate the promotive role of glucose metabolism, we utilized the hexokinase inhibitors 3-Bromopyruvic acid (BrPA) and 2-Deoxy-D-glucose (2DG)[23] to block glucose utilization during differentiation (Fig. 1d). Apoptosis levels showed no significant toxicity under the concentrations of BrPA and 2DG used in our experiments (Supplementary Fig. 1d). Importantly, both inhibitors remarkably reduced the proportion of CXCR4 and SOX17 double-positive cells in a dose-dependent manner (Fig. 1e). And the suppression of glucose metabolism was further confirmed by the decreased level of acetyl-

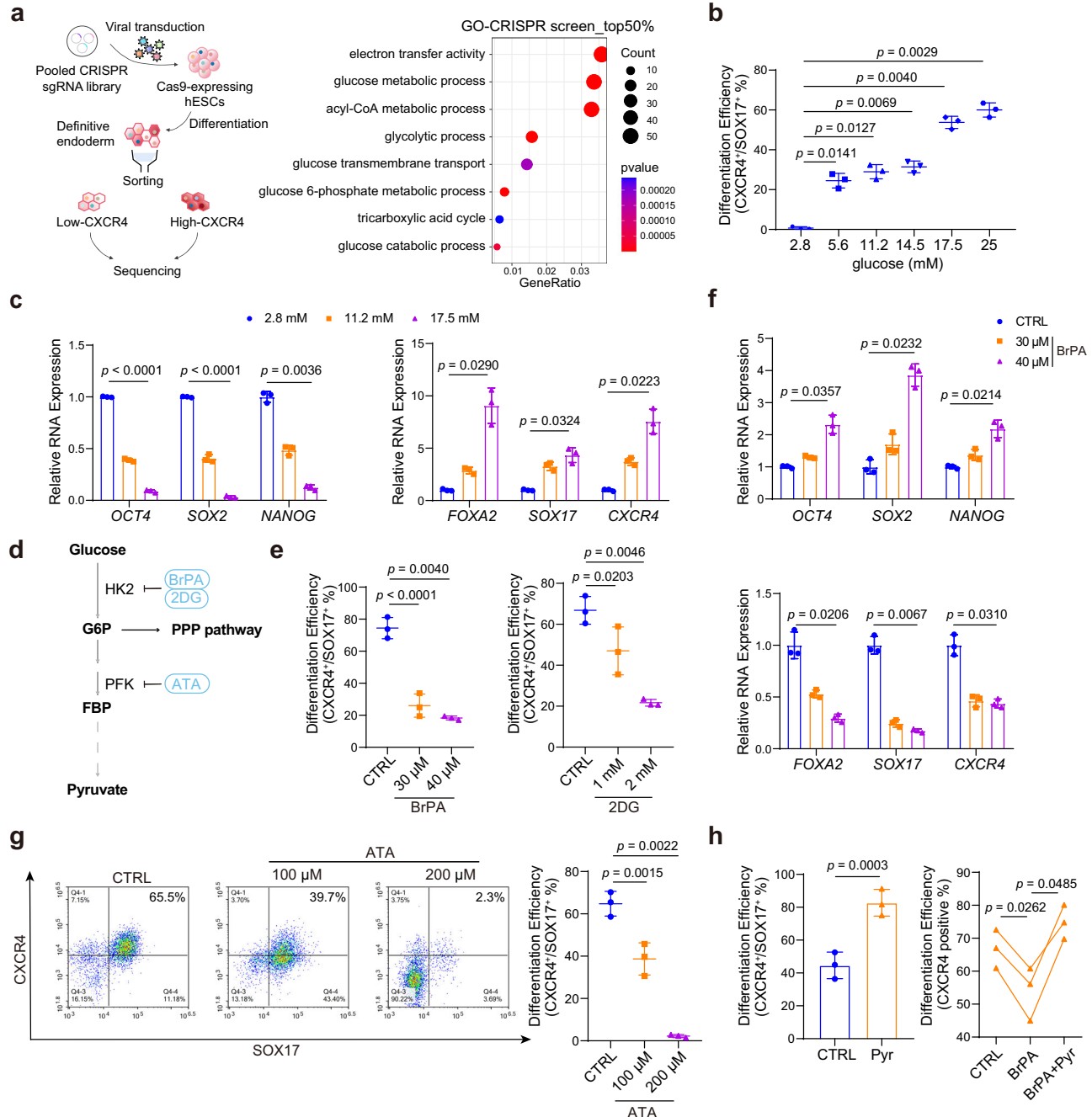

**Fig. 1 | Glucose metabolism deficiency impairs DE differentiation. a** Schematic overview of the CRISPR screen (left) and GO enrichment analysis of the top 50% genes differentially enriched in the top 30% CXCR4-positive versus bottom 30% CXCR4-negative cells (right). **b** Flow cytometric analysis of SOX17 and CXCR4 showing the DE differentiation efficiency treated with different glucose concentrations, in HUES8 ($n = 3$ independent experiments). **c** mRNA levels of DE marker genes *FOXA2, SOX17, CXCR4* and ESC marker genes *OCT4, SOX2, NANOG* under different glucose concentrations, in HUES8 ($n = 3$ independent experiments). **d** Schematic illustration of glucose utilization and early metabolism, and the inhibitors with their targets used in the following experiments. **e** Flow cytometric analysis of SOX17 and CXCR4 showing the DE differentiation efficiency treated with BrPA and 2DG. The data shown about BrPA is from HUES8, and 2DG data from PGP1 (n = 3 independent experiments). **f** mRNA levels of *FOXA2, SOX17, CXCR4* and *OCT4,* *SOX2, NANOG* treated with 30 μM and 40 μM BrPA, respectively, measured by qRT-PCR, in HUES8 ($n = 3$ independent experiments). **g** Flow cytometric analysis of SOX17 and CXCR4 showing the DE differentiation efficiency treated with 100 μM and 200 μM ATA, respectively, in HUES8 ($n = 3$ independent experiments). **h** Flow cytometric analysis showing the DE differentiation efficiency upon BrPA treatment or control, with or without exogenous pyruvate, in HUES8 ($n = 3$ independent experiments). Each point represents an individual replicate. Statistics were calculated using one-way ANOVA, followed by Dunnett's multiple comparisons test in (**b** and **h**) (right). Two-way ANOVA for (**c** and **f**) followed by Tukey's multiple comparisons test. One-sided hypergeometric test with multiple comparison adjustments was used in (**a**) (right). For comparisons between two groups, two-tailed paired *t* tests were applied directly. All data are presented as mean ± SD. Source data are provided as a Source Data file.

CoA, a key intermediate of this pathway (Supplementary Fig. 1e). Consistent with the flow cytometric result, we found endoderm marker genes (*CXCR4, SOX17* and *FOXA2*) were down-regulated and pluripotent genes (*OCT4, SOX2* and *NANOG*) were up-regulated along with

the increased concentration of inhibitors (Fig. 1f and Supplementary Fig. 1f). Together, these results highlight the crucial role of glucose metabolism in DE differentiation, which is positively correlated with the differentiation efficiency.

## Glucose-pyruvate pathway promotes DE differentiation

To further validate the downstream of glucose metabolism, we used phosphofructokinase (PFK) inhibitor Aurintricarboxylic acid (ATA)[24], which blocks phosphofructokinase and possibly facilitates the PPP pathway to generate reduced nicotinamide adenine dinucleotide phosphate (NADPH) and ribose-5-phosphate[25], crucial for regulating cellular oxidative stress and producing metabolic intermediates needed for cell proliferation (Fig. 1d). Interestingly, addition of ATA significantly decreased the protein levels of DE markers SOX17 and CXCR4 (Fig. 1g). Consistently, quantitative real-time PCR (qRT-PCR) results confirmed that DE differentiation was markedly inhibited upon ATA treatment (Supplementary Fig. 1g), indicating that the fructose-1,6-bisphosphate (FBP)-pyruvate metabolism, rather than PPP, functions in DE differentiation.

In addition, we supplemented the metabolite, pyruvate, the downstream product of glucose utilization, followed by FBP and evaluated whether pyruvate is involved in DE differentiation. The data showed that pyruvate can effectively promote differentiation. Most importantly, supplementation with pyruvate can rescue the impaired DE differentiation caused by BrPA treatment (Fig. 1h). Taken together, we concluded that the glucose-pyruvate pathway is required for human DE differentiation.

To further investigate the effects of BrPA treatment on differentiation potential toward other germ layers, we performed embryoid body (EB) formation assays. We found that BrPA treatment did not significantly affect EB size (Supplementary Fig. 2a). qRT-PCR analysis revealed that BrPA treatment slightly impaired expressions of mesendodermal markers (*EOMES*, *MIXL1*, *T*) and terminal endodermal markers (*SOX17*, *FOXA2*, *CXCR4*). In addition, ectodermal differentiation was also disrupted (Supplementary Fig. 2b). In addition, we performed directed differentiation toward ectoderm and mesoderm lineages under BrPA treatment. For ectoderm differentiation, neuroectodermal markers (*NES, PAX6, SOX1*) were examined on days 3 and 7. These markers remained largely unchanged at day 3 but were significantly reduced by day 7 upon BrPA treatment (Supplementary Fig. 2c). For mesoderm differentiation, we used a previously reported cardiomyocyte differentiation protocol[26]. The expression of cardiac-specific markers (*TNNT2, GATA4, MYH6*) was impaired in BrPA-treated cells (Supplementary Fig. 2d), indicating impaired differentiation toward the mesodermal cardiac lineage. Interestingly, similar to what we observed in DE differentiation, the reduced ectodermal differentiation in EBs could be also rescued by pyruvate or glutamine supplement (Supplementary Fig. 2e). These results together suggest that the glucose-pyruvate pathway is crucial for proper lineage differentiation of human ESCs.

## Pyruvate entering the TCA cycle supports DE differentiation

The direction of glucose metabolic flux is largely determined by whether pyruvate is converted into lactate by the lactate dehydrogenase (LDH) complex, or into acetyl-CoA by the pyruvate dehydrogenase (PDH) complex (Fig. 2a)[27,28]. In our RNA-seq data, we found glycolysis-related genes were predominantly downregulated, whereas the TCA cycle-related genes were largely upregulated during DE differentiation (Supplementary Fig. 3a, b). To determine which direction is functional and required for DE differentiation, we attempted to interrupt the lactate production and the TCA cycle, respectively.

We first used LDH inhibitors oxamic acid sodium (OXA)[29] and galloflavin (GF)[30] to block lactate production and thus promote pyruvate entry into the TCA cycle. We found that OXA and GF treatment increased the proportion of CXCR4 and SOX17 double-positive cells (Fig. 2b), and the analysis of DE-related gene expressions under both treatments also showed that OXA and GF promoted cell differentiation. (Supplementary Fig. 3c). As expected, OXA treatment indeed inhibited lactate production while promoting pyruvate entry into the TCA cycle (Fig. 2d, e), and GF also increased the acetyl-CoA levels

(Supplementary Fig. 3d). Therefore, we concluded that lactate production is harmful for DE differentiation.

Meanwhile, we also used the mitochondrial pyruvate carrier (MPC) inhibitor UK5099 to impede pyruvate entry into mitochondria, and the pyruvate dehydrogenase kinase (PDK) inhibitor sodium dichloroacetate (DCA) to enhance PDH activity[31,32]. The results showed that UK5099 significantly decreased the differentiation efficiency (Fig. 2c). Consistent with this finding, RNA expression analysis also indicated that UK5099 inhibited the DE-related gene expression (Supplementary Fig. 3c). And the levels of lactate and acetyl-CoA also indicated that UK5099 suppressed pyruvate conversion to acetyl-CoA, thus promoting lactate production and decreasing the acetyl-CoA content (Fig. 2d, e). Importantly, treatment with PDK inhibitor DCA, which facilitates the pyruvate utilization, also led to the anticipated differentiation-promoting effects (Supplementary Fig. 3e). Notably, acetyl-CoA levels also were elevated under DCA treatments, indicating an increase in TCA cycle flux (Supplementary Fig. 3d). We also evaluated the effect of glutamine, a TCA cycle metabolic precursor. The data indicate that glutamine significantly promotes differentiation (Fig. 2f). More importantly, supplementation with glutamine effectively rescues DE differentiation following treatment with BrPA or 2DG (Fig. 2g). Taken together, these results indicated that the metabolism switch from the glycolysis to TCA cycle could effectively facilitate the DE differentiation.

## TGFβ promotes pyruvate metabolism during DE differentiation

Next, we wanted to investigate the exact regulation mechanism of the metabolism switches. The LDH and PDH complexes are major regulators of pyruvate metabolism flux. We first examined the gene expression changes of these complexes during DE differentiation. qPCR results revealed that PDH-related genes were obviously upregulated during differentiation (Fig. 3a). We also assessed the genes encoding mitochondrial pyruvate transporters, *MPC1* and *MPC2*, exhibiting notable increases (Supplementary Fig. 4a). Therefore, we hypothesized that a specific signal during differentiation controlled the upregulation of PDH-related genes, thereby mediating the metabolic switching. Following this hypothesis, we focused on the TGFβ and WNT signaling pathways, which are the major regulatory pathways of DE differentiation[33]. We examined the chromatin immunoprecipitation followed by sequencing (ChIP-seq) data of SMAD2/3 and β-catenin[34], the downstream transcription factor of the TGFβ pathway and WNT pathway, respectively. Interestingly, SMAD2/3 significantly bound to the PDHB with two distinct peaks, while β-catenin showed no bindings (Fig. 3b). We also confirmed this conclusion by ChIP-qPCR (Fig. 3c). In contrast, SMAD2/3 does not show significant binding peaks on *LDHA* or *LDHB*, correlating with the expression changes of LDH-related genes (Supplementary Fig. 4b).

To further confirm the regulatory role of the TGFβ signaling pathway on PDHB expression, we activated the TGFβ signal pathway by activator Activin A or overexpressing SMAD3. Western blot analysis showed that TGFβ activation significantly increased PDHB expression (Fig. 3d, e). Similar results were observed in HEK293T cells, and both RNA and protein levels of PDHB were significantly upregulated following TGFβ pathway activation (Fig. 3f, g). Overall, these data indicate that the TGFβ signaling pathway can activate *PDHB* expression during DE differentiation, thereby mediating the metabolic model switch.

## PDHB is critical for DE differentiation

The PDH complex is essential for converting pyruvate to acetyl-CoA, which is a critical step for sustaining TCA cycle flux[35]. To further validate that pyruvate entry into the TCA cycle is important for DE differentiation, we employed CRISPR-Cas9 technology to target *PDHB*, a critical component of PDH complex, based on the HUES8 cell line (Supplementary Fig. 5a). A previous study demonstrated that PDHB depletion markedly reduces PDH activity and disrupts metabolic

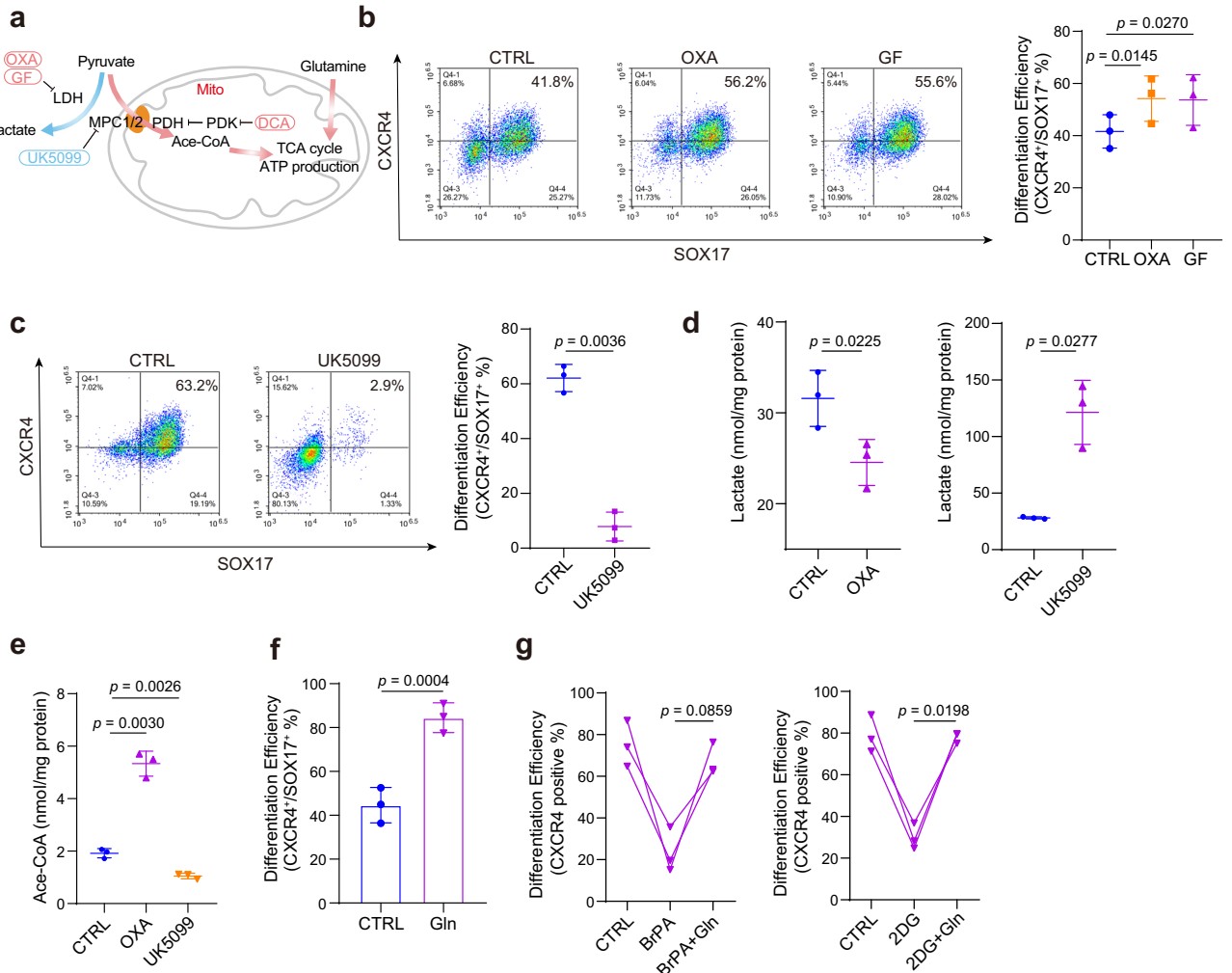

**Fig. 2 | DE differentiation favors the TCA cycle rather than lactate production. a** Schematic illustration of pyruvate metabolism flux (TCA cycle, lactate production) and the inhibitors with targets used in the following experiments. **b, c** Flow cytometric analysis of SOX17 and CXCR4 showing the DE differentiation efficiency treated with OXA, GF (**b**) and UK5099 (**c**), in HUES8 ($n = 3$ independent experiments). **d, e** Total lactate (**d**) and acetyl-CoA (**e**) content in DE cells treated with OXA and UK5099 inhibitors, in HUES8 ($n = 3$ independent experiments). **f** Flow cytometric analysis of SOX17 and CXCR4 showing the DE differentiation efficiency

supplemented by exogenous glutamine, in HUES8 ($n = 3$ independent experiments). **g** Flow cytometric analysis of CXCR4 showing the DE differentiation efficiency upon BrPA or 2DG treatment with or without exogenous glutamine, in HUES8 ($n = 3$ independent experiments). Each point represents an individual replicate. Statistics were calculated using one-way ANOVA, followed by Dunnett's multiple comparisons test in **g**. For comparisons between two groups, two-tailed paired $t$ tests were applied directly. All data are presented as mean ± SD. Source data are provided as a Source Data file.

patterns[36]. Although we generated ten heterozygous clones out of 53 genotyped clones, we did not obtain any homozygous PDHB-knockout human ESCs (Supplementary Fig. 5b), suggesting PDHB might be essential for ESC survival.

Nevertheless, heterozygous cell lines displayed a significant decrease of PDHB expression (Supplementary Fig. 5c) and metabolic alterations, including reduced acetyl-CoA levels and increased lactate levels (Fig. 4a). The pluripotency of PDHB[+/-] cells was not influenced by decreased PDHB and acetyl-CoA (Supplementary Fig. 5d). Therefore, we proceeded with the heterozygous lines for further experiments. We first assessed whether this metabolic switch affected differentiation efficiency using flow cytometry and an immunofluorescence assay. Results indicated that PDHB[+/-] cells had a reduced proportion of CXCR4 and SOX17 double-positive cells and fewer FOXA2-positive cells than wild-type cells (Fig. 4b and Supplementary Fig. 5e), suggesting that the TCA cycle inhibition impairs endodermal differentiation. Then we performed RNA-seq using differentiated PDHB[+/-] cells. The results revealed profound changes in gene expression (Supplementary Fig. 5f). We also applied single-cell flux estimation analysis

(scFEA) to investigate the metabolic flux changes in PDHB[+/-] cells[37]. Consistently, metabolites related to TCA cycle intermediates (e.g., acetyl-CoA, oxaloacetate, Fumarate) were enriched in wild-type cells. And metabolites related to glycolysis (e.g., lactate) were remarkably enriched in PDHB[+/-] cells, suggesting PDHB heterozygous knockout blocked TCA cycle metabolism fluxes and upregulated glycolysis (Fig. 4c). Furthermore, the gene-set enrichment analysis (GSEA) highlighted the enrichment of DE-related genes in wild-type cells and ESC-related genes in PDHB[+/-] cells (Fig. 4d), reinforcing that differentiation efficiency was impaired in PDHB[+/-] cells. Overall, these findings demonstrate that the PDHB-mediated pyruvate utilization in the TCA cycle supports endodermal differentiation, and its inhibition adversely impairs the differentiation efficiency.

We also performed EB differentiation assays on the PDHB[+/-] cell line. The results showed that PDHB heterozygous knockout attenuated the downregulation of pluripotency markers (OCT4, SOX2, NANOG) and impaired endodermal and ectodermal differentiation. Specifically, it led to decreased expression of endodermal markers (SOX17, FOXA2, CXCR4) and ectodermal genes (NES, PAX6).

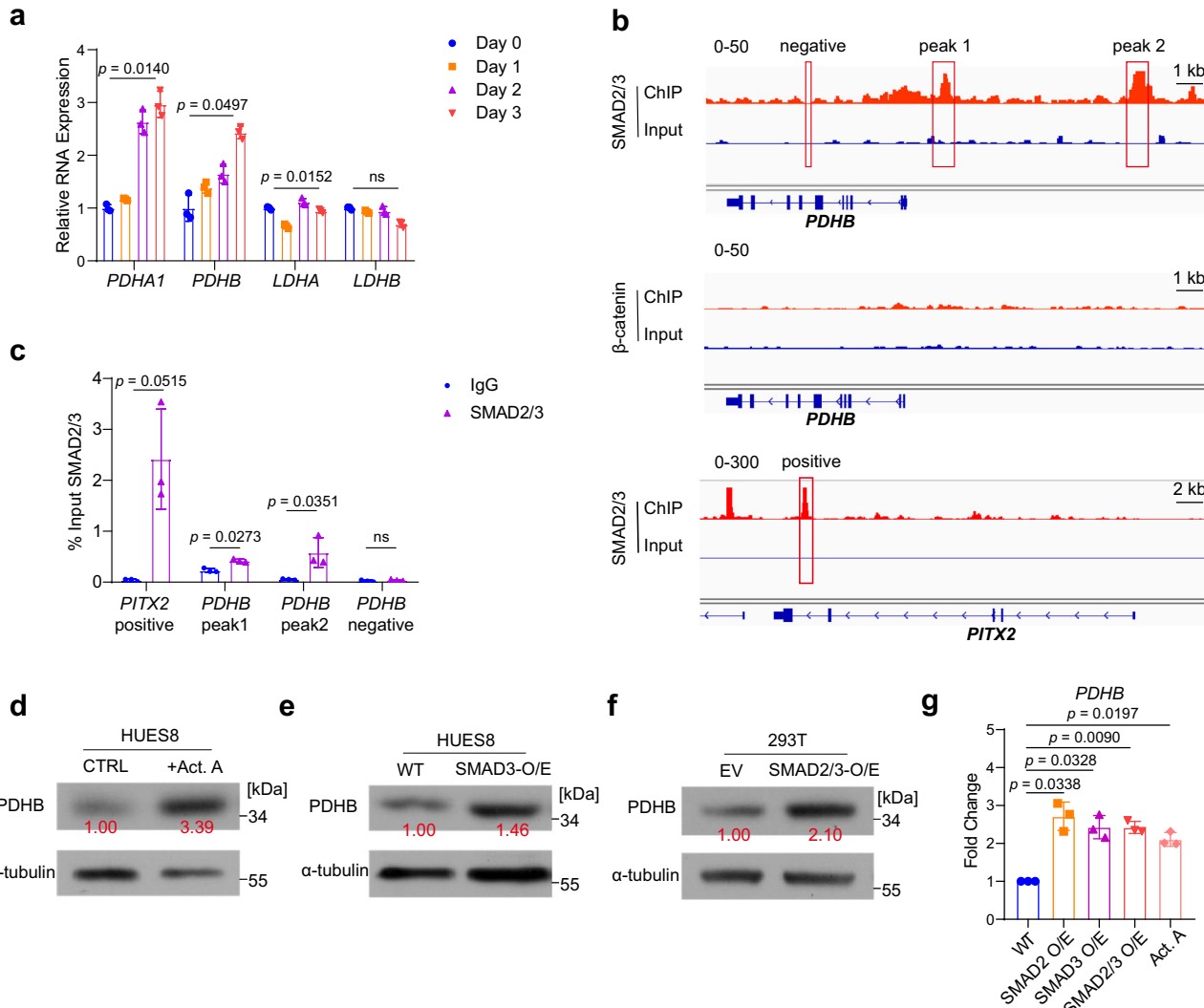

**Fig. 3 | TGFβ/SMAD2/3 regulates the expression of metabolic gene *PDHB*.**
**a** mRNA levels of *PDHA1*, *PDHB*, *LDHA*, *LDHB* during the DE differentiation process, measured by qRT-PCR, in HUES8 ($n = 3$ independent experiments). **b** Integrative Genomics Viewer (IGV) plots illustrate the binding levels of transcription factors SMAD2/3 and β-catenin on PDHB. **c** ChIP-qPCR analysis of SMAD2/3 binding to the PDHB locus ($n = 3$ independent experiments). **d** Protein levels of PDHB upon Activin A treatment in HUES8 cells. The values shown in the figure represent the results after normalization of the target protein to the loading control, followed by further standardization ($n = 3$ independent experiments). **e** Protein levels of PDHB upon overexpressing SMAD3 in HUES8 cells. The values shown in the figure represent the results after normalization of the target protein to the loading control, followed by

further standardization ($n = 3$ independent experiments). **f** Protein levels of PDHB upon overexpressing SMAD2/3 in 293 T cells. The values shown in the figure represent the results after normalization of the target protein to the loading control, followed by further standardization ($n = 3$ independent experiments). **g** mRNA levels of *PDHB* in 293 T cells treated with SMAD2-OE, SMAD3-OE, SMAD2/3-OE, Activin A, measured by qRT-PCR. Data were normalized to WT. ($n = 3$ independent experiments). Each point represents an individual replicate. Statistics were calculated using one-way ANOVA, followed by Dunnett's multiple comparisons test in (**g**). Two-way ANOVA for (**a**) followed by Tukey's multiple comparisons test. For comparisons between two groups, two-tailed paired *t* tests were applied directly. All data are presented as mean ± SD. Source data are provided as a Source Data file.

Interestingly, we found that the expression of early mesendodermal stage markers (*EOMES, MIXL1, T*) are increased (Fig. 4e). Given that pluripotent stem cells first transition through a mesendodermal stage, marked by the expression of *EOMES, MIXL1*, and *T* (Brachyury), before further differentiating into either mesoderm or DE[38], we analyzed gene expression dynamics at three time points (Day 3, 5, and 9) during EB differentiation. We observed that PDHB[+/−] cells exhibited a delayed downregulation of pluripotency genes and a postponed peak in mesendodermal gene expression: mesendodermal markers peaked around Day 5 in wild-type cells, while the PDHB[+/−] cells reached this stage around Day 9 or later (Supplementary Fig. 5g). This data suggests that PDHB depletion impairs both the exit from pluripotency and the initial mesendodermal differentiation. Notably, the upregulation of DE markers was continually delayed in PDHB[+/−] cells.

The DE differentiation represented an early stage of pancreatic lineage commitment. We further examined whether impaired DE differentiation affected subsequent pancreatic development based on our published pancreatic differentiation protocol[39]. At both the early pancreatic progenitor (PP1) and late pancreatic progenitor (PP2) stages, BrPA treated and PDHB[+/−] cells exhibited markedly reduced RNA expression of key pancreatic genes, including *FOXA2*, *NKX6-1*, *NKX2-2*, and *PDX1* (Supplementary Fig. 5h), which was further confirmed by the immunofluorescence analysis (Supplementary Fig. 5i). Overall, these results suggest that metabolic alterations caused by BrPA treatment or PDHB deficiency not only impair three germ layers differentiation but also disrupt the progression of pancreatic differentiation.

We next established PDHB-overexpressing ESC lines and confirmed efficient overexpression at both the RNA and protein levels

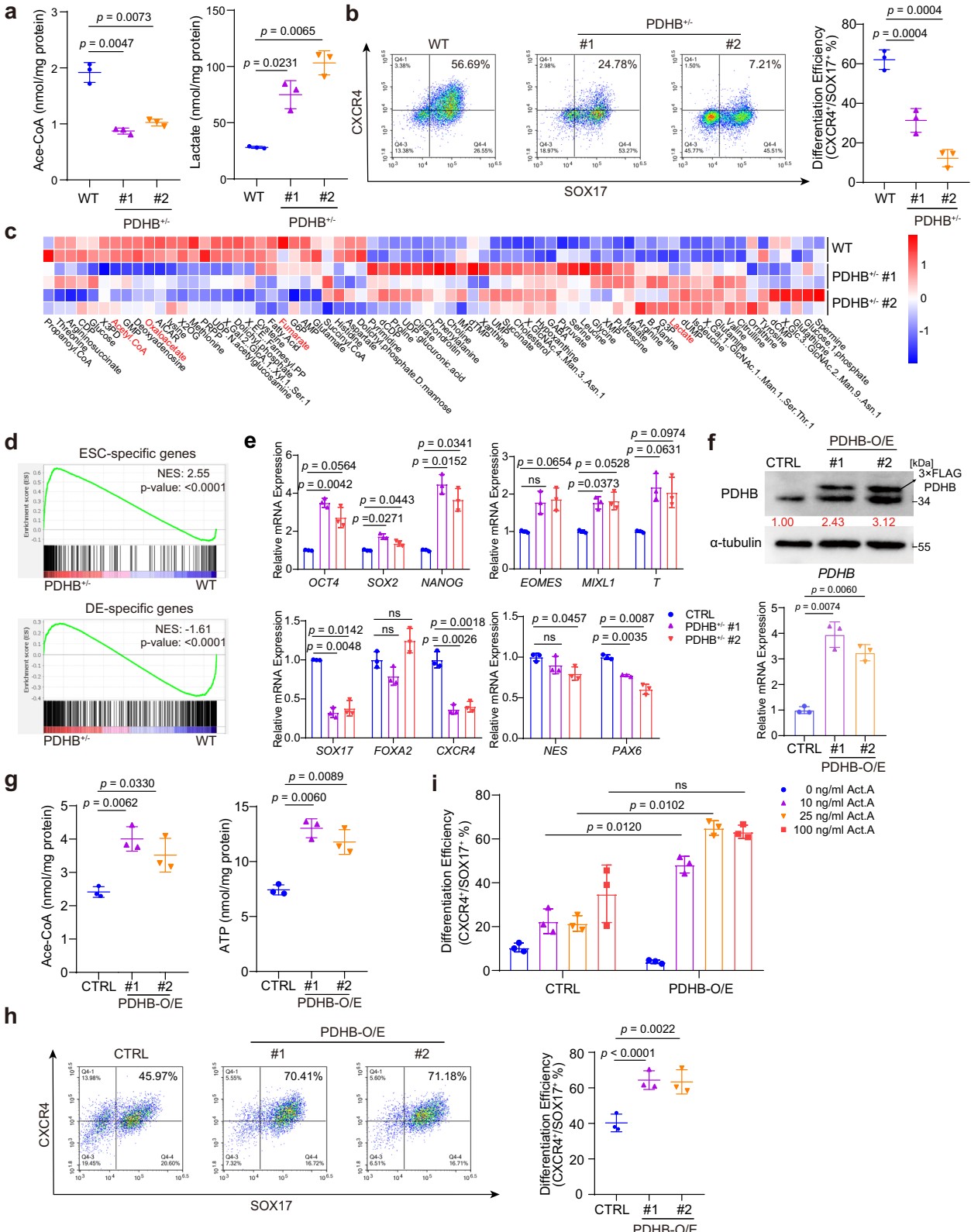

(Fig. 4f). PDHB upregulation enhanced the conversion of pyruvate to acetyl-CoA, thereby stimulating the TCA cycle activity and ATP production (Fig. 4g). To further assess the metabolic consequences, [U-13C]-glucose tracing revealed increased glucose-derived labeling in key TCA intermediates, including citrate, aconitate, succinate, and succinyl-CoA (Supplementary Fig. 6a, b and Supplementary Data 1), consistent with an overall metabolic shift toward enhanced oxidative metabolism. More importantly, PDHB overexpression significantly improved DE differentiation efficiency (Fig. 4h). Moreover, under reduced Activin A concentrations, PDHB overexpression further promoted DE differentiation (Fig. 4i), suggesting that PDHB activity becomes rate-limiting when signaling input is weakened. Collectively, these findings support that PDHB drives a metabolic shift toward increased TCA cycle flux, thereby enhancing DE differentiation.

**Fig. 4 | PDHB can change metabolism switch and affect differentiation. a** Total acetyl-CoA and lactate content in PDHB$^{+/-}$ DE cells, in HUES8 ($n = 3$ independent experiments). **b** Flow cytometric analysis of SOX17 and CXCR4 showing the DE differentiation efficiency in PDHB$^{+/-}$ DE cells, in HUES8 ($n = 3$ independent experiments). **c** Global profile of predicted metabolic fluxes. **d** GSEA profile of DE-specific genes and ESC-specific genes in WT and PDHB$^{+/-}$ cells. **e** mRNA expression levels of pluripotency markers (*OCT4, SOX2* and *NANOG*) and lineage-specific markers (endoderm: *SOX17, FOXA2, CXCR4*; mesendoderm: *EOMES, MIXL1, T*; ectoderm: *NES, PAX6*) in CTRL and PDHB$^{+/-}$ EBs on day 9 ($n = 3$ independent experiments). **f** the RNA and protein levels of *PDHB*-O/E cells. The values shown in the figure represent the results after normalization of the target protein to the loading control, followed by further standardization, in HUES8 (n = 3 independent experiments). **g** ATP and acetyl-CoA content in the PDHB-overexpressing cell line ($n = 3$ independent experiments). **h** Flow cytometric analysis of SOX17 and CXCR4 showing the DE differentiation efficiency in PDHB-O/E cells, in HUES8 ($n = 3$ independent experiments). **i** DE differentiation efficiency of CTRL and PDHB-O/E cells under different Activin A concentrations, measured by flow cytometric analysis. The commonly-used concentration is 100 ng/ml ($n = 3$ independent experiments). Each point represents an individual replicate. Two-way ANOVA for (**e** and **i**), followed by Tukey's multiple comparisons test. The permutation-based, one-sided test was performed in (**d**). For comparisons between two groups, two-tailed paired *t* tests were applied directly. All data are presented as mean ± SD. Source data are provided as a Source Data file.

## ATP and BAF complex play crucial roles in DE differentiation

Next, we wanted to investigate how such a metabolism switch regulated DE differentiation. Various metabolic intermediates could affect protein post-translational modifications or chromatin structure by acting as substrates or cofactors of epigenetic enzymes[8]. Thus, we supplemented SAM, α-KG, acetate, citrate, or ATP to determine which metabolite could rescue DE differentiation upon BrPA treatment and PDHB heterozygous knockout. Interestingly, we found that SAM, α-KG, acetate and citrate showed no apparent rescue effect (Supplementary Fig. 7a), whereas ATP significantly restores endodermal differentiation efficiency under both conditions (Fig. 5a). Consistently, we observed a significant ATP levels reduction in both BrPA-treated and PDHB$^{+/-}$ cells (Fig. 5b). Moreover, ATP content correlated with differentiation efficiency across different inhibitor treatments: 2DG, ATA and UK5099 (inhibit DE differentiation) decreased ATP levels, while OXA, GF, and DCA (accelerate DE differentiation) increased ATP levels (Supplementary Fig. 7b). Moreover, upon treated with another PDH inhibitor CPI613[40], DE differentiation was blocked along with decreased ATP content, while the exogenous supplement of ATP could rescue the differentiation defect (Supplementary Fig. 7c), consistent with the observations from PDHB$^{+/-}$ cells. Overall, these findings suggest that the impact of metabolic switch on differentiation efficiency is mainly mediated by the changes in ATP content.

Then we analyzed the difference of gene expression pattern between PDHB$^{+/-}$ and wild-type cells to further investigate the interface between ATP level and DE differentiation. GO enrichment analysis showed that metabolic-related gene sets, such as acetyl-CoA metabolism and differentiation-associated gene sets like the WNT signaling pathway, cell fate determination, and endoderm formation were enriched in differentially expressed genes, as expected. Interestingly, we found that chromatin accessibility associated terms were also enriched, such as DNA-binding transcription activator activity and regulation of DNA binding (Supplementary Fig. 7d). Literature further highlights the crucial roles of ATP-dependent chromatin remodeling complexes during the exit from pluripotency, such as Brahma-associated factor (BAF) complexes and the NURD complex[41]. Therefore, we hypothesized that metabolic switching and ATP content might influence the function of ATP-dependent chromatin remodeling complexes, thereby affecting differentiation efficiency.

To test this hypothesis, we first examined whether the ATP-dependent chromatin remodeling complex and which complex regulated endoderm differentiation. We used specific inhibitors of the BAF complex (BRM014 targeting core subunits BRM/BRG1, and BI7273 targeting BRD7/9) and NURD complex (BPK-25) during DE differentiation. The results showed that the inhibition of BAF complex activity significantly impaired differentiation efficiency, whereas inhibition of the NURD complex did not have any obvious impact (Fig. 5c), suggesting that the differentiation inhibition was more likely mediated by the BAF complex. The BAF complex consists of two mutually exclusive core subunits, BRM and BRG1[42]. We further examined the expression of these core subunits during differentiation and found that BRG1 is predominantly expressed and progressively upregulated during differentiation, while BRM is scarcely expressed (Fig. 5d and Supplementary Fig. 7e). Next, we attempted to generate BRG1 knockout ESCs. Given that BRG1 knockdown disrupts pluripotency maintenance in human[43,44], we employed the dTAG-induced degradation system for cell line construction[45]. Western blotting confirmed complete BRG1 degradation within 8 hours of dTAG induction in these two HUES8-derived ESC clones (Fig. 5e). Consistent with the inhibitor treatment, BRG1 loss during DE differentiation significantly impaired endoderm formation (Fig. 5f). Taken together, these data indicate that the BAF complex, especially the BRG1-centered BAF complex, is necessary for endodermal differentiation.

To further establish that ATP functions upstream of BRG1, we inhibited BAF activity during ATP supplementation. Inhibition of the BAF complex effectively abolished the rescuing effect of ATP on DE differentiation (Fig. 5g). Similarly, the decreased RNA levels of DE marker genes *FOXA2, SOX17, CXCR4* due to blocked glucose utilization and mitochondrial pyruvate metabolism could be also rescued by ATP supplement, then inhibited by BAF inhibitors (Supplementary Fig. 8a). Moreover, the result that ATP supplementation failed to restore DE differentiation defect in BRG1-depletion cells (Fig. 5h) further confirmed that BRG1 is required for ATP-mediated enhancement of differentiation.

In addition, we employed BRM014, a specific inhibitor of the ATPase activity of BRM/BRG1[46]. The results showed that exogenous ATP supplementation failed to promote DE differentiation when BRG1's ATPase activity was inhibited (Supplementary Fig. 8b, c), indicating ATP exerted its effect on differentiation through the ATPase function of BRG1. More importantly, overexpression of wild-type or ATPase-dead (K798R[47,48]) BRG1 in dTAG-induced BRG1-knockout cells revealed that only wild-type BRG1 restored DE differentiation, whereas the ATPase-dead mutant failed to rescue (Fig. 5i). Together, these results demonstrate that BRG1 regulates DE differentiation through its ATPase activity.

## Metabolic remodeling during human embryonic development

To further investigate whether similar mechanisms including metabolic reprogramming and the ATP-BAF regulatory axis operate in vivo, we analyzed a publicly available single-cell RNA-seq dataset of 3D-cultured human pre-gastrulation embryos[49], encompassing multiple cell types present during the gastrulation stage, including the embryonic disc, amnion, yolk sac and the primitive streak anlage (PSA)[50] (Supplementary Fig. 9a). Based on the data, we examined the expression patterns of genes involved in glycolysis, the TCA cycle, oxidative phosphorylation, and the BAF chromatin remodeling complex across different embryonic tissues. Our analysis revealed that during the developmental transition from the inner cell mass (ICM) to epiblast (EPI) and subsequently to PSA-EPI, glycolytic gene expression (including canonical glycolytic genes such as *GAPDH, GPI, HK2*, and *LDHA*) was markedly decreased, whereas genes related to the TCA cycle and oxidative phosphorylation were significantly upregulated (Supplementary Fig. 9a, b). These results indicate a clear metabolic shift toward mitochondrial oxidative metabolism during this stage. Furthermore, genes associated with the BAF complex, such as *BRG, BRD7, BRD9* and *ARID1A*, also exhibited a moderate

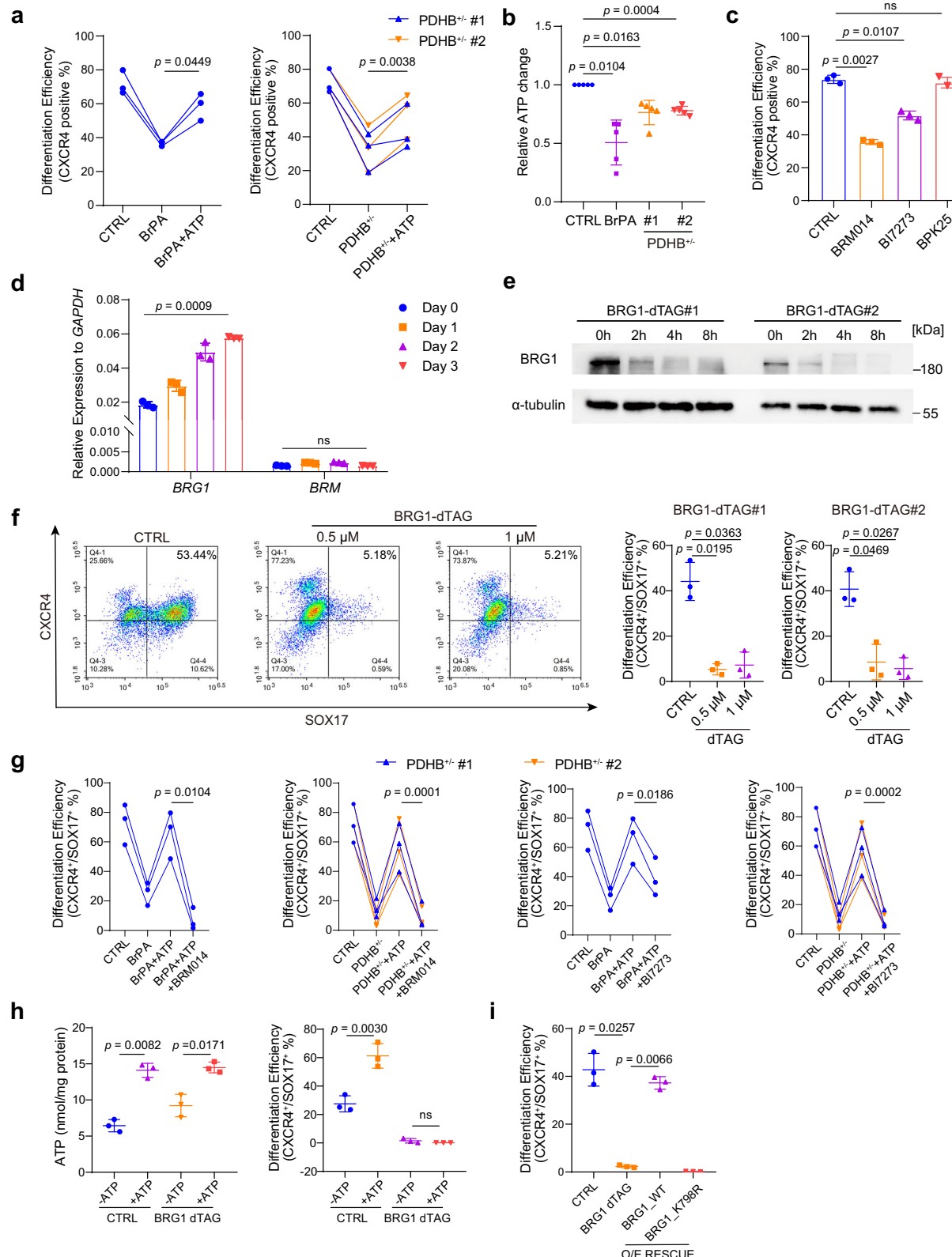

increase in expression throughout the ICM-EPI-PSA-EPI differentiation process (Supplementary Fig. 9c). Together, these findings suggest that the metabolic and epigenetic changes described in our study also occur during normal human embryonic development. Notably, such a metabolic transition was not observed in lineages differentiating toward extraembryonic fates (Supplementary Fig. 9a, b).

## PDHB promotes chromatin accessibility in DE genes by BAF

To confirm whether the function of the BAF complex is affected upon inhibiting mitochondrial pyruvate metabolism in the PDHB[+/-] cells, we performed the ATAC-seq (Assay for Transposase-Accessible Chromatin with high throughput sequencing) to assess changes of chromatin accessibility. We first confirmed the data quality evidenced by the nice reproducibility among different samples (Fig. 6a and Supplementary

**Fig. 5 | Alterations in glucose-pyruvate metabolism affect DE differentiation through ATP concentration and the activity of BAF complex. a** Flow cytometric analysis of CXCR4 showing the DE differentiation efficiency upon BrPA or in PDHB[+/-] cells with or without exogenous ATP, in HUES8 ($n = 3$ independent experiments). **b** Total ATP content in DE cells treated with BrPA and in PDHB[+/-] DE cells ($n = 5$ independent experiments). **c** Flow cytometric analysis of CXCR4 showing the DE differentiation efficiency treated with BRM014, BI7273 or BPK25, in HUES8 ($n = 3$ independent experiments). **d** mRNA levels of *BRG1* and *BRM* during DE differentiation, measured by qRT-PCR, in HUES8 ($n = 3$ independent experiments). **e** Protein levels of dTAG induced BRG1 degradation in 0-8 hours. ($n = 3$ independent experiments). **f** Flow cytometric analysis of SOX17 and CXCR4 showing the DE differentiation efficiency treated with dTAG induced BRG1 degradation ($n = 3$ independent experiments). **g** Flow cytometric analysis of SOX17 and

CXCR4 showing the DE differentiation efficiency treated with BRM014 or BI7273, together with exogenous ATP ($n = 3$ independent experiments). **h** Measurement of ATP content and flow cytometric analysis of SOX17 and CXCR4 expression in control and dTAG-induced BRG1 degradation groups, with or without ATP supplementation ($n = 3$ independent experiments). **i** Flow cytometric analysis of SOX17 and CXCR4 showing the DE differentiation efficiency of dTAG-induced BRG1 degradation cell line, rescue with WT or K798R mutation BRG1 ($n = 3$ independent experiments). Each point represents an individual replicate. Statistics were calculated using one-way ANOVA, followed by Dunnett's multiple comparisons test in (**a**–**c**, **i**). Two-way ANOVA for (**d**, **h**) followed by Tukey's multiple comparisons test. For comparisons between two groups, two-tailed paired *t* tests were applied directly. All data are presented as mean ± SD. Source data are provided as a Source Data file.

Fig. 10a). Differential peaks analysis revealed that PDHB depletion significantly impacted the overall chromatin accessibility. Interestingly, most of the differential peaks showed decreased accessibility following metabolic suppression (Fig. 6a). The finding further indicated that the ATP-dependent chromatin remodeling complex BAF was functionally impaired in PDHB[+/-] cells. We next performed GSEA of the top 500 upregulated and downregulated genes identified by ATAC-seq, and the results confirmed that changes in chromatin accessibility were correlated with RNA-seq data (Supplementary Fig. 10b). Moreover, we conducted GO enrichment analysis on genes with reduced chromatin accessibility in PDHB[+/−]. The results also showed significant enrichment in gene sets associated with pancreas development, endoderm development, and the WNT signaling pathway (Fig. 6b), consistent with the inhibited DE differentiation we observed previously.

Subsequently, to investigate the relationship between the chromatin accessibility changes and BAF complex function, we reanalyzed BRG1 ChIP-seq data[51] and overlapped the genes with downregulated ATAC-seq signal and BRG1-binding signal (Fig. 6c). The Venn diagram showed that the majority of genes with decreased chromatin accessibility also bind to the BRG1 protein. Notably, most of these genes were upregulated during DE differentiation and were enriched in DE cells compared with ESCs (Fig. 6d, e). And the up-regulated genes in day 5 also enriched the terms related to the endoderm differentiation, including the stem cell differentiation and canonical WNT signal pathway (Fig. 6f). Moreover, DE-related transcription factors, including *TCF4* and *FOXI1*, showed marked reductions in chromatin accessibility and gene expression following PDHB depletion (Fig. 6h). Taken together, the functional impairment of BAF complex leads to more closed chromatin accessibility of BAF binding DE-related genes, consequently disrupting the cell differentiation.

Several studies have demonstrated that the BAF complex primarily remodels the chromatin environment of enhancers[52–54]. To investigate whether BRG1 exerts its function through the regulation of enhancer accessibility in DE differentiation, we further analyzed the distribution of BRG1-binding peaks across different genomic elements. The results revealed that BRG1 preferentially binds to intergenic and intronic regions, accounting for approximately 74.9% of the total binding sites (Supplementary Fig. 10c), while it also bound to promoter regions for about 10.76%. Then we compared chromatin accessibility of enhancer and active transcription start site (TSS) regions between wild-type and PDHB-depleted cells. The data indicated that only the chromatin accessibility at enhancers was significantly reduced in the PDHB-deficient cells, while there were no notable changes in TSS regions (Fig. 6g). As examples, BRG1 primarily binds the non-promoter regions of *TCF4* and *FOXI1*, which were typically marked by two enhancer-associated histone modifications, H3K27ac and H3K4me1 (Fig. 6h). To further confirm the BRG1 binding was dependent on the metabolic pattern, we performed ChIP-qPCR on these non-promoter regions of DE-related genes in wild-type and PDHB-deficient cells. The results revealed that PDHB depletion led to

an obvious decrease in BRG1 enrichment at DE-related gene loci, such as *FOXI1*, *TCF4*, *SMAD3* and *SOX17* (Fig. 6i and Supplementary Fig. 10d), concomitant with the reduced chromatin accessibility observed in ATAC-seq. These findings indicated that PDHB and ATP production facilitated the BRG1/BAF complex to bind with enhancers and carries out its function of facilitating chromatin opening.

## Discussion

Our present study reveals that human DE differentiation undergoes significant glucose metabolic remodeling, including a decrease in lactate production and an increase in pyruvate-TCA cycle activity; furthermore, such glucose metabolic alterations indeed impact DE differentiation. In detail, DE differentiation favors the high TCA cycle metabolic profile, which is driven by the upregulation of *PDHB* expression activated by the TGFβ signal. Such metabolic changes in turns influence BRG1/BAF complex function primarily through alterations in ATP levels, thereby regulating DE differentiation. Overall, our findings have established the link between glucose metabolic switch and chromatin accessibility via ATP and the ATP-dependent BRG1 complex (Fig. 7).

Several studies have revealed a close link between metabolism and epigenetic regulation. Typically, the metabolic intermediates can influence epigenomic status through histone modifications[55,56]. The histone undergoes various post-translational modifications such as acetylation, methylation, and phosphorylation, which ultimately impact the chromatin environment and gene expression. Notably, the addition or erasure of these modifications not only requires specific catalytic enzymes but also rely on metabolic intermediates. For instance, all known acetyltransferases require acetyl-CoA as the acetyl donor[57]; cellular methylation processes also depend on metabolic intermediates such as SAM as methyl donors or TCA cycle intermediate α-KG as important cofactors[58]. In this study, we observed significant rescue of DE differentiation only with ATP under both inhibition conditions of blocking glucose metabolism. As an ATP-dependent chromatin remodeler, further rescue experiments support that the BRG1/BAF complex contributes to DE differentiation modulated by glucose metabolism. Indeed, the BAF complex can epigenetically regulate pluripotency and lineage differentiation. For instance, BRM, one core subunit of the BAF complex, safeguards the canalization of cardiac mesoderm differentiation from mouse ESCs. Mechanistically, the loss of BRM suppresses the chromatin accessibility of cardiac enhancers and prevents the binding of neural suppressor REST with BAF complexes[59]. During mesoderm and endoderm differentiation, a recent study has found that lineage-related transcription factor EOMES cooperates with BAF complex and SMAD2/3 to accelerate mesoderm and endoderm differentiation[60]. In addition, ATPase activity assays and restriction enzyme accessibility assays (REAA) demonstrate that ATP contents can affect the nucleosome-remodeling activity of BAF complex[61]. Consistent with these reports, we found that chromatin accessibility decreased in BRG1 binding regions and enhancers when blocking glucose utilization and ATP production

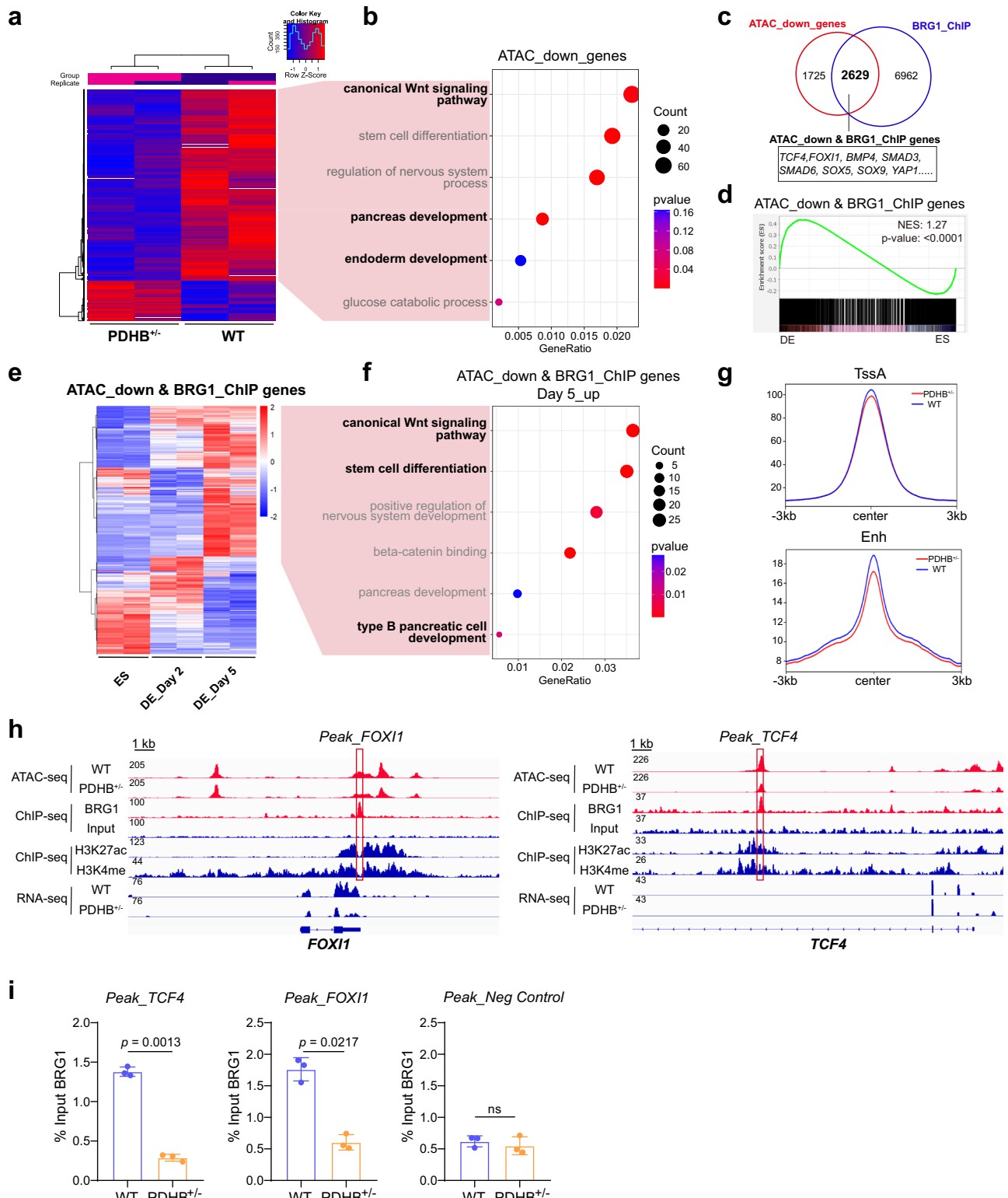

(Fig. 6c–g and Supplementary Fig. 10d). We thus proposed a new perspective that ATP could establish the connection with the epigenome through the ATP-dependent chromatin remodeling complex BAF in human ESC fate determination.

In our study, we applied external ATP to regulate the cellular energy state. Since ATP hardly cross the cell membrane freely, whether it can enter the cell and the possible mechanisms need further dissection. Several earlier studies have demonstrated that ATP can be both released from and taken up by cells[62,63]. Recent works using non-hydrolyzable fluorescent ATP (NHF-ATP) tracing methods have shown that multiple cell types, including human lung cancer cell A549 and human non-small cell lung cancer (NSCLC) cells, can internalize extracellular ATP via macropinocytosis, leading to an increase in intracellular ATP levels[64–66]. These findings provide strong evidence for the internalization of extracellular ATP and suggest that cellular energy status can be modulated through exogenous ATP supplementation. In our experiments, we directly measured intracellular ATP levels and confirmed that ATP supplementation successfully restored ATP concentrations in DE cells

**Fig. 6 | The chromatin accessibility of BRG1-binding regions becomes lower in PDHB$^{+/-}$ DE cells. a** Heatmap showing the differentially chromatin accessibility peaks by ATAC-seq between WT and PDHB$^{+/-}$ DE cells. Red indicates higher chromatin accessibility, while blue indicates lower chromatin accessibility. **b** GO enrichment analysis of reduced chromatin accessibility genes (related to **a**). **c** Venn diagram between genes with downregulated ATAC-seq signal genes (ATAC_down_genes) and genes with BRG1-binding (BRG1_ChIP). **d** GSEA profile of 2629 overlapped genes (ATAC_down & BRG1_ChIP genes) in Fig. 6c. **e** Heatmap of the 2629 overlapped ATAC_down & BRG1_ChIP genes (related to **c**) in ESC and DE (Day 2, Day 5) cells. Red denotes higher expression, while blue denotes lower expression. **f** GO enrichment analysis of the upregulated ATAC_down & BRG1_ChIP genes during DE differentiation. **g** ATAC-seq signal coverage of WT and PDHB$^{+/-}$ cells at enhancers (Enh) and active TSS (TssA) regions. **h** IGV plots illustrate the levels of BRG1, H3K27ac, H3K4me (by ChIP-seq), and changes of chromatin accessibility (by ATAC-seq) and expression (by RNA-seq) between WT and PDHB$^{+/-}$ DE cells. **i** ChIP-qPCR of BRG1 on DE-related gene enhancers (*FOXI1, TCF4*). The primers were designed to detect the red-lined peaks in (**h**) (*n* = 3 independent experiments). Each point represents an individual replicate. One-sided hypergeometric test with multiple comparison adjustments was used in (**b**, **f**). The permutation-based, one-sided test was performed in (**d**). For comparisons between two groups, two-tailed paired *t* tests were applied directly. All data are presented as mean ± SD. Source data are provided as a Source Data file.

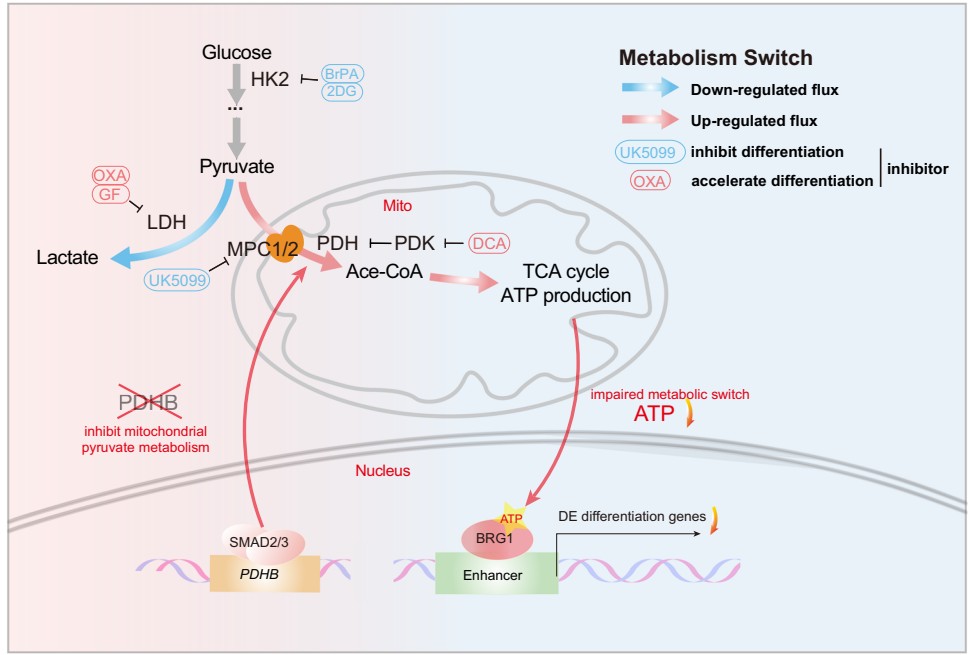

**Fig. 7 | A model about glucose metabolism switch affecting human endoderm differentiation by ATP and BRG1/BAF complex.** TGFβ/SMAD2/3 could bind to *PDHB* loci and regulate *PDHB* expression. The PDH complex controls the mitochondrial pyruvate metabolism. Inhibition of mitochondrial pyruvate metabolism during DE differentiation impairs the BAF complex function by reducing ATP production. Altered ATP contents then lower the chromatin accessibility of differentiation-related genes via BAF, leading to impaired DE differentiation.

(Supplementary Fig. 8b, c). This restoration occurred both under normal conditions and when BRG1 ATPase activity was inhibited, further supporting that extracellular ATP can influence intracellular ATP levels and, consequently, impact cellular processes dependent on ATP availability.

Interestingly, we also observed the reduced acetyl-CoA in BrPA and 2DG treated cells besides ATP levels. The acetyl-CoA is not only related with TCA cycle and ATP production, but also an important donor of histone acetylation modification. Why doesn't acetyl-CoA affect DE differentiation via ATP levels instead of influencing histone acetylation states? Moussaieff A and colleagues found that differentiation cells had lower acetyl-CoA levels and histone acetylation (H3K27ac and pan-H3ac)[1]. In our study, BrPA treatment led to reduced acetyl-CoA and H3K9/K27ac (Supplementary Fig. 11a). Based on this result, the BrPA would accelerate the pluripotency exit and differentiation process. However, the experiments data was conflicted with this hypothesis. It means that the acetyl-CoA exerts function rely on ATP production rather than epigenetic modification.

As the key downstream transcriptional factor of the TGFβ superfamily, SMAD2/3 controls many cell functions, such as cell proliferation and cell fate determination. Particularly, the TGFβ signal pathway participates in the PSC pluripotency maintenance and germ layers differentiation. In human ESCs, TGFβ-activated SMAD2/3 is necessary for maintaining undifferentiated markers such as *NANOG*[67]. Mechanistically, the SMAD3 co-occupies pluripotency-related genes with OCT4, finally maintaining undifferentiation state[68]. SMAD2/3 can also co-operate with WNT effector β-catenin to promote differentiation when NANOG and PI3K signals are low[69]. Interestingly, there are a few recent studies indicating other regulatory layers also involved in SMAD2/3-mediated cell fate determination, such as long noncoding RNAs[70] and RNA methylation[71]. Here we found that SMAD2/3 could bind with the TCA cycle-related gene *PDHB* (Fig. 3b), and TGFβ activation could induce the expression of PDHB (Fig. 3d–g), thus mediating the upregulation of mitochondrial pyruvate metabolism and TCA cycle. Consistently, a recent study reported that TGFβ could control the mitochondrial respiration by regulating target gene Mss51[72]. Another report indicated that Smad3 could bind with the promoter region of NOX4, thereby induced ROS production and inflammation[73]. These studies together with our present work demonstrate that TGFβ could regulate lineage differentiation via a metabolic manner.

In addition, glucose, not only is a major energy source, but also serves in glycolysis branch pathways including serine synthesis pathway (SSP), hexosamine biosynthesis pathway (HBP), and PPP pathway to provide key precursors for nucleotide biosynthesis and reducing power for various anabolic processes[25]. Our study focuses on the downstream pyruvate metabolism, not excluding the importance of other pathways, albeit our results suggest that PPP seems not to mainly contribute to DE differentiation. In addition, due to the limited access to embryo materials, we did not provide a direct in vivo perturbation

assay to demonstrate glucose utilization and mitochondrial pyruvate metabolism is indeed functional in early endodermal development. We noticed a recent study described about glucose uptake during the gastrulation in mouse embryo. Cao and colleagues found that glucose actively participates in the gastrulation with two stage-specific waves of glucose metabolism. The first wave occurs by the hexosamine biosynthetic pathway (HBP) to drive epiblast fate acquisition, and the second wave mainly uses glycolysis to mediate mesoderm migration and lateral expansion[74]. Further studies on the mapping and dissection of glucose metabolism flux during human in vivo endoderm specification would be of great value.

In summary, we have dissected the functional role of glucose-pyruvate-TCA cycle in human DE differentiation. TGFβ could transcriptionally activate *PDHB* gene expression and reprogram the glucose metabolic flux, contributing to endoderm differentiation. Higher activity of glucose-pyruvate-TCA cycle makes more ATP and enhances the function of ATP-dependent chromatin remodeling complexes BAF, thereby increasing the accessibility of differentiation-associated genes in chromatin level. Our study not only demonstrates that glucose metabolic processes are closely linked to and functional in human early differentiation, but also establishes a new link among TGFβ, metabolic remodeling, epigenetic regulation, and cell fate determination.

## Methods

### Cell culture

Human embryonic stem cell line HUES8 and induced pluripotent stem cell line PGP1 were cultured on Matrigel-coated plates (Corning, Cat#354277) with mTeSR1 medium (STEMCELL Technologies, Cat#85850) at 37 °C with 5% CO$_2$. Subculturing was performed every 4 days using Accutase (STEMCELL Technologies, Cat#07922), and the culture medium was changed daily. Human embryonic kidney cell line 293 T (HEK293T) was grown on tissue culture-treated plates to ensure cell adherence, in DMEM mixed with 10% fetal bovine serum (FBS, Gibco, Cat#10100147) and 1 x penicillin/streptomycin (Gibco, Cat#10378016). Passaging was carried out using 0.05% trypsin (Gibco, Cat#25300120) when cells reached 90–100% confluence.

The chemical small molecules used in this study were as follows: 3-Bromopyruvic acid (BrPA), at concentrations of 30-40 μM (APExBio, Cat#B7922); 2-Deoxy-D-glucose (2DG), at concentrations of 1-2 mM (APExBio, Cat#B1027); ATA, at concentrations of 100–200 μM (Sigma, Cat#A1896); Oxamate (OXA), at a concentration of 5 mM (MCE, Cat#HY-W013032A); Galloflavin (GF), at a concentration of 50 μM (APExBio, Cat#B5716); UK5099, at a concentration of 4 μM (MCE, Cat#HY-15475); Dichloroacetate (DCA), at a concentration of 10 mM (APExBio, Cat#B7174); BRM/BRG1 Inhibitor-1, at a concentration of 25 nM (MCE, Cat#HY-119374); BI-7273, at a concentration of 10 μM (Selleck, Cat#S8179); and BPK-25, at a concentration of 5 μM (MCE, Cat#HY-141550). Notably, the concentrations of these chemicals used in this study had a minimal impact on cell survival.

### Definitive endoderm (DE) differentiation

For the induction of DE, experiments were conducted using 24-well cell culture plates coated with growth factor-reduced Matrigel (Corning, Cat#354230). The initial cell seeding density was 100,000 cells per well. After cultivating the cells in mTeSR1 for one day, the DE differentiation medium was switched for the next three days. The DE differentiation medium was based on DMEM (Gibco, Cat#C11995500BT) and supplemented with 100 ng/mL Activin A (PeproTech, Cat#120-14 P), 1 x penicillin/streptomycin (Gibco, Cat#10378016), and 0.2% bovine serum albumin (Yeasen Biotechnology, Cat#36104ES25).

### CRISPR-screen

Human ESCs were infected by metabolism-related lentiviral library which contains Cas9 cassette. Then the cells were selected by 2 μg/ml puromycin for 2 days. After the selection, we passaged the cells to DE differentiation medium, which contained 100 ng/mL Activin A, 1 x penicillin/streptomycin, and 0.2% fatty acid-free bovine serum albumin. Then we tested the differentiation efficiency with APC-conjugated mouse anti-human CXCR4 antibody. Genomic DNA was isolated from the top 30% positive and bottom 30% negative cells, and the sgRNA sequences were amplified by PCR and then sequenced.

### Gene overexpression constructs

The coding sequence (CDS) of *PDHB* was subcloned into the doxycycline-inducible pCW lentiviral vector. Lentiviral particles were generated by co-transfecting HEK293T cells with the pCW-PDHB construct together with the packaging plasmids psPAX2 and pMD2.G The viral supernatant was collected and transfected into HUES8 cells. Stable PDHB-overexpressing cells were obtained following selection with 2 μg/mL puromycin for about 7 days. Wild-type and ATPase-mutant BRG1 were overexpressed in HUES8 cells using the PiggyBac system.

### [U-$^{13}$C]-glucose Metabolic flux analysis

Cells were cultured in glucose-free medium supplemented with 17.5 mM [U-$^{13}$C]-glucose (MCE, HY-B0389A). After incubation, metabolites were extracted with 80% methanol at −80 °C and analyzed in the National Protein Science Technology Center at Tsinghua University using a Dionex Ultimate 3000 UPLC coupled to a TSQ Quantiva Ultra triple-quadrupole mass spectrometer (Thermo Fisher). Data were acquired in selected reaction monitoring (SRM) under positive/negative ion switching mode. TCA cycle intermediates (citrate, α-ketoglutarate, succinate, etc.) were detected for $^{13}$C isotopologue analysis, and quantified using TraceFinder 3.2 (Thermo, USA).

### RNA preparation and qRT-PCR

Cells were first washed with Dulbecco's phosphate-buffered saline (DPBS, Gibco, Cat#C14190500BT) and then harvested using lysis buffer. Total RNA was extracted using the HiPure Total RNA Mini Kit (Magen, Cat#R4111-03) according to the manufacturer's protocol. Qualified total RNA, assessed with NanoDrop 2000 spectrophotometer (Thermo Fisher), was reverse-transcribed into complementary DNA (cDNA) using the ABScript II RT Master Mix (ABclonal, Cat#RK20402) following the manufacturer's guidelines. qRT-PCR was conducted using a C1000 Touch Thermal Cycler (Bio-Rad) with a 384-well plate format. The reaction included 2x SYBR Green qPCR Master Mix (Bimake, Cat#B21203).

### Flow cytometry

Cultured cells were detached using TrypLE (Gibco, Cat#12604013) at 37 °C for 2 minutes. After detachment, cells were washed twice with DPBS containing 2% FBS and collected by centrifugation at 300 g for 3 minutes at 4 °C. Cells were then resuspended in DPBS with 2% FBS and incubated with APC-conjugated mouse anti-human CD184 antibody (1: 200, BD, Cat#555976) for 30 minutes at 4 °C in the dark. Following incubation, the cells were washed twice with cold DPBS containing 2% FBS. The cells were fixed and permeabilized using Fix/Perm Buffer (BD Biosciences, Cat#562574) for 1 hour at 4 °C. After fixation, cells were washed twice with Perm Buffer (BD Biosciences, Cat#562574). Cells were then incubated with Alexa Fluor 488-conjugated mouse anti-human SOX17 antibody (1: 200, BD Biosciences, Cat#562205) for 45 minutes at 4 °C. After staining, cells were washed and resuspended in 200 μL of DPBS. The stained cells were analyzed using the CytoFLEX flow cytometer (Beckman).

### Immunofluorescence staining

Cells were initially fixed with 4% paraformaldehyde for 15 minutes at room temperature. After fixation, they were washed three times with DPBS and subsequently blocked and permeabilized with a solution containing 10% donkey serum and 0.3% Triton X-100 in DPBS for 1 hour

at room temperature. Primary antibodies were then applied in a solution of DPBS with 0.3% Triton X-100 and 10% donkey serum and incubated overnight at 4 °C. The following day, the cells were washed three times with DPBS and then incubated with secondary antibodies in the same blocking solution for 1-2 hours at room temperature. After washing with DPBS, the cells were counterstained with 4,6-diamidino-2-phenylindole (DAPI, Sigma, Cat#10236276001) for 10 minutes at room temperature. The primary antibodies used included: rabbit anti-FOXA2 (1: 200, HUABIO, China, Cat#ET1703-76), anti-PDX1 (1: 200, R&D, Cat#AF2419), anti-NKX6-1 (1: 200, ABclonal, Cat#A20419), and anti-NKX6-1 (1: 200, Cell Signaling Technology, Cat#54551).

## Western blot
Cells were washed with DPBS, dissociated using TrypLE, and collected by centrifugation at $300 \times g$ for 3 minutes at 4 °C. Whole cell extracts were prepared using RIPA buffer (Beyotime, Cat#P0013C) supplemented with a protease inhibitor cocktail (Roche, Cat#4693116001). The protein concentrations were determined using a BCA protein assay kit (Thermo Fisher Scientific, Cat#A53225). Approximately 20 μg of protein was separated by SDS-PAGE (10% gel) and transferred to a nitrocellulose membrane (Millipore, Cat#Z746010). The membrane was then blocked with TBST, which contained 5% BSA, incubated overnight at 4 °C with primary antibodies, and washed with TBST for 30 minutes. HRP-conjugated secondary antibodies were applied for 1 hour at room temperature, followed by washing and treatment with Immobilon Western Chemiluminescent HRP Substrate (Millipore, Cat#WBKLS0500) for visualization. The antibodies for western blot used in the present work include: α-Tubulin (1: 3000, Proteintech, Cat#11224-1-AP), PDHB Polyclonal antibody (1: 2000, Proteintech, Cat#14744-1-AP), and SMARCA4/BRG1 Polyclonal antibody (1: 5000, Proteintech, Cat#21634-1-AP). The original western blots were shown in the Supplementary material.

## Generation of knockout cell lines
To generate *PDHB*-knockout ESCs, CRISPR/Cas9 genome editing technology was employed. The guide RNA sequence of *PDHB* (5′-CTAGAGAATGAATTGATGTA-3′) was designed by the online CRISPR design tool (http://crispr.mit.edu/). Then the sgRNA oligos were annealed and ligated into the pX459 expression vector, and then the construct was co-electroporated with a GFP vector (approximately 1/10 of the pX459 amount) into ESCs using the P3 Primary Cell 4D-Nucleofector X Kit. The transfected cells were sorted based on GFP expression via flow cytometry and subsequently plated into a 96-well plate. Genotyping of the resulting colonies was performed by Sanger sequencing, and PDHB expression levels were verified by Western blot analysis.

For BRG1-dTAG cells, we used CRISPR/Cas9-mediated homology-directed recombination HDR to knock in a sequence encoding 3×Flag-FKBP-T2A-Puromycin at the C-terminus of BRG1. The pX459 vector carrying the guide RNA (5′-GCTGCTACCCGTTACTGCTA-3′) and the HDR template was transfected into HUES8 cells using Lipofectamine (Thermo Fisher, Cat#STEM00001). After puromycin selection, surviving cells were single-cell sorted, and successful knock-in clones were verified by PCR.

## Measurement of acetyl-CoA, L-lactate and ATP levels
Cells were washed with DPBS, dissociated with TrypLE, incubated at 37 °C for 2 minutes, resuspended in DMEM, and collected by centrifugation at $300 \times g$ for 3 minutes at 4 °C. The quantification of acetyl-CoA was performed by the Acetyl CoA Assay Kit (sigma, Cat#MAK039), the quantification of L-Lactate was performed by the L-Lactate Assay Kit (abcam, Cat#ab65330), and the quantification of ATP was performed by the ATP Assay Kit (Beyotime, Cat#S0026B). For normalization of cell mass, protein concentration was measured with the Pierce BCA Protein Assay Kit (Thermo Fisher, Cat#23225).

## RNA-Seq and Data analysis
The RNA-seq data were processed based on a previous pipeline[76]: reads were aligned to the human genome (hg38) by bowtie2 (2.3.4.3). Secondary alignments were performed to ensure that each read was uniquely mapped to a single genomic location. Subsequently, read counts were obtained with featureCounts (1.6.3). RNA expression levels were normalized as TPM (transcripts per million). Differential expression analysis was conducted with DESeq2, and GSEA was performed on the GSEA software to identify specifically enriched pathways.

The sequencing data in this study have been uploaded to the Gene Expression Omnibus (GEO) database under accession codes GSE285132, GSE285133, GSE285135. The public data used in this study was list as follow: BRG1 ChIP-seq data from GSE182842[51], H3K27ac ChIP-seq data and H3K4me ChIP-seq data from GSE54471[77], β-catenin ChIP-seq data from GSE99202[34], the DE RNA-seq data from GSE168625[18].

## Single-cell flux estimation analysis (scFEA)
The scFEA was used to predict the metabolite changes based on the single-cell or bulk RNA-seq[37]. The detailed protocol of scFEA can be found on GitHub (https://github.com/changwn/scFEA). In brief, TPM from RNA-seq was the input of the scFEA package. And analysis was performed in default parameters. Finally, the results were visualized using the pheatmap package.

## ChIP-seq and ATAC-seq
For each sample, $1 \times 10^7$ cells were crosslinked with 1% formaldehyde for 10 minutes and quenched with 0.125 M glycine for 5 minutes. Cells were washed with cold DPBS, collected by scraping, and resuspended in Cell Lysis Buffer (10 mM Tris-HCl, pH 8.0, 140 mM NaCl, 0.2% NP-40) with 1x protease inhibitor. The lysis was sonicated to generate 200−500 bp fragments. Chromatin was immunoprecipitated overnight with 5 μg SMAD2/3 antibody (R&D, Cat#AF3797) or Protein A/G Dynabeads. After extensive washing, DNA was eluted and reverse crosslinked at 67 °C for 5 h. The purified DNA was sent for ChIP-seq library construction.

Cells were prepared for ATAC-seq library construction using the Hyperactive ATAC-Seq Library Prep Kit for Illumina (Vazyme, Cat#TD711-01) and the TruePrep Index Kit V2 for Illumina (Vazyme, Cat#TD202). The ATAC-seq data were processed as follows: all reads were aligned using Bowtie2 (2.3.4.3), and duplicates were removed. Subsequent data visualization followed a similar approach to ChIP-seq (reported in our previous works)[76], where deeptools (3.3.0) was used to generate bigWig (bw) files, which were then utilized to produce heatmaps and average profile plots[78–80].

## Statistics and reproducibility
All experiments were conducted independently with at least three biological replicates if not specifically mentioned. Data were analyzed using GraphPad Prism 8 and were presented as means ± standard deviation (SD). For figures with three or more groups, statistics were calculated using one-way ANOVA, followed by Dunnett's multiple comparisons test. Two-way ANOVA was used when two variables were involved, followed by Tukey's multiple comparisons test. For comparisons between two groups, two-tailed paired *t* tests were applied directly. Statistical significance is indicated in each figure. No statistical method was used to predetermine sample size; no data were excluded from the analyses; the experiments were not randomized; and investigators were not blinded to allocation during experiments and outcome assessment.

## Ethics
The human ESCs and iPSCs reported in this study have been approved by the Biomedical Ethics Committee of Wuhan University (approval

number: WHU-LFMD-IRB2024026). The established human ESC line HUES8, acquired from the Harvard Stem Cell Institute, was used in this study[75].

## Data availability
Original data generated in this study have been deposited at GEO under the accession codes GSE285132, GSE285133, and GSE285135. The CRISPR-screening data generated in this study have been deposited in the NCBI database under accession code PRJNA1390417. Published data sets analyzed in the manuscript are available in the GEO database under accession codes GSE182842, GSE54471, GSE99202, and GSE168625. This paper does not report any original code. Source data are provided in this paper.

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

## Acknowledgements

We thank Profs. Guoliang Qing at Wuhan University and Donghui Zhang at Hubei University for providing reagents and the insightful discussion. We would like to thank Xiafei Zhang and other laboratory members for technical help and discussion. This work is supported by the National Natural Science Foundation of China (No. 32350019), the Natural Science Foundation of Hubei Province (Innovation Group project, 2024AFA018), the Modern Microbial Fermentation and Cell Engineering Program at Cangzhou Normal University (cxtdl2303), and the Fundamental Research Funds for the Central Universities in China (2042022dx0003).

## Author contributions

W.J. conceived and supervised the project together with J.Y.; and designed the experiment together with L.M., Y.Y., and J.L; Y.Y and X.L. performed the screening experiment; J.L., Y.Y. and L.M. performed the inhibitor tests; L.M. finished the mechanistic dissect; L.M. analyzed the RNA-seq, ATAC-seq and ChIP-seq data with help from C.Y.; L.Z. analyzed the scRNA-seq data from the database. L.M. drafted the manuscript, and

W.J., L.M., and J.Y. finalized the manuscript. All authors contributed to and approved the final manuscript.

## Competing interests

The authors declare no competing interests.
