## [Transparent Peer Review file · Nature Communications]

TGF β -activated PDHB promotes mitochondrial pyruvate metabolism and contributes to human endoderm differentiation via ATP-dependent BRG1

Corresponding Author: Professor Wei Jiang

Version 0:

Reviewer comments:

Reviewer #1

(Remarks to the Author)

This manuscript by Meng et al. provides compelling evidence that glucose metabolism, specifically, mitochondrial pyruvate flux governed by TGF β -induced PDHB expression, plays a pivotal role in definitive endoderm (DE) differentiation from human pluripotent stem cells (hPSCs). The authors demonstrate that ATP generated through this metabolic reprogramming enhances the activity of the BRG1/BAF chromatin remodeling complex, thereby promoting chromatin accessibility at DE-related gene loci. Through a comprehensive combination of CRISPR screening, small molecule perturbations, metabolite supplementation, RNA-seq, ATAC-seq, and ChIP-seq, the study establishes a mechanistic link between extracellular signaling, cellular metabolism, epigenetic remodeling, and cell fate determination. The study employs innovative approaches to uncover a previously underappreciated role of TGF β -PDHB signaling in directing glucose metabolism toward mitochondrial pyruvate oxidation. However, several concerns should be addressed prior to publication:

1. While the study implicates BRG1 as a downstream effector of ATP-mediated chromatin remodeling, direct evidence of its ATPase activity modulation is lacking. Incorporating experiments with a dominant-negative or ATPase-dead BRG1 mutant would strengthen the causal relationship.
2. It remains unclear whether PDHB overexpression or activation is sufficient to enhance DE differentiation. Addressing this would clarify whether PDHB acts as a limiting factor in this process.
3. Some figures, particularly Fig. 6 depicting ATAC-seq and BRG1 ChIP-seq overlap, are densely annotated. Simplifying these visuals and including schematic models would improve accessibility and reader comprehension.
4. Terminology such as "mainstream glucose metabolism" versus "PPP" should be clearly defined earlier in the manuscript to avoid ambiguity and improve clarity.

Reviewer #2

(Remarks to the Author)

The authors revealed a metabolic switch with decreased lactate production and increased mitochondrial pyruvate metabolism, which is controlled by TGF β -activated PDHB, is necessary for the definitive endoderm differentiation from human pluripotent stem cells. The experiments data, especially the pyruvate rescue and ATP rescue, are convincing and impressive, and the following are the additional points that can be addressed.

In line 55-56 in the introduction, the glucose metabolism is also required for the cell type specification of TE during blastocyst development.

For figure 1, the authors can add one or two sentence to explain the rationale of using CXCR4 as the readout of DE differentiation in their CRISPR screening.

For figure 1, whether glucose metabolism inhibition leads to compensation of other lineage differentiation, such as ectoderm

and mesoderm, or it generally inhibits differentiation and pluripotency exit in this 2D differentiation system. It was discussed in the embryoid body section but not here. Also, the toxicity effect data upon BrPA/2DG treatment was mentioned but not shown.

For figure 2, the pyruvate and glutamine rescue experiments are interesting and convincing in elucidating the role of glucose metabolism in DE differentiation. Can they also rescue the slight changes of ectoderm differentiation in Fig 2I?

In line 276-279, the authors should better explain the difference of mesoendoderm and definitive endoderm to make it coherent through the article

Can the authors use a PDH inhibitor to show ATP decrease and reduced DE differentiation, which can be rescued by ATP, in addition to the general glycolysis inhibitor?

For fig 5, the ATP rescue experiments are impressive. Can supplemented ATP be permeable to the cell membrane to change the intracellular ATP level, and is there other reports or this is the first time to show ATP can participate in the ATP-dependent chromatin remodeling processes. These points can be at least discussed in the discussion section.

Some of the grammar should be polished such as :

Line 139 To further validate the promotional role

Line 144 evident by the decreased acetyl-CoA

Line 169 Pyruvate entering TCA cycle facilitates DE differentiation

TCA cycle should be the TCA cycle.

Reviewer #3

(Remarks to the Author)

Summary

Meng et al. investigates how glucose metabolism regulates endoderm differentiation from human iPS cells. The authors demonstrate that mitochondrial pyruvate metabolism is required for ATP production, which in turn supports the chromatin remodeling activity of the BRG1-containing BAF complex. They further link this metabolic switch to TGF β signaling through transcriptional regulation of PDHB by SMAD2/3. Loss of PDHB impairs DE differentiation, which can be partially rescued by exogenous ATP, and this rescue is blocked by BAF complex inhibition. These findings are supported by extensive molecular analyses including CRISPR screens, metabolite measurements, ATAC-seq, ChIP-seq, and RNA-seq.

While the study makes a compelling case for the involvement of mitochondrial metabolism in endoderm fate commitment, several major conceptual, technical, and interpretational limitations lessen its current suitability for publication in Nature Communications.

Conceptual Impact and Novelty / Overall Big Picture Issues

The manuscript brings forward a conceptually appealing model that connects growth factor signaling (TGF β), metabolic state (pyruvate flux \rightarrow ATP), and chromatin remodeling (BAF complex) to orchestrate cell fate decisions in human-iPS cells. The novelty resides in linking ATP-dependent chromatin remodeling to a specific metabolic switch regulated by PDHB, as a direct target of SMAD2/3. This integration of metabolism and epigenetics during germ layer specification is timely and of interest to a broad audience.

However, several gaps undermine the model's completeness and generalizability:

- The central mechanism (ATP \rightarrow BAF) is not fully validated, and alternative ATP-dependent processes are not addressed.
- The metabolic reprogramming is inferred, not directly measured via flux analysis.
- Findings are derived from a single cell line, limiting generalizability.
- The in vivo relevance is not addressed at all.

Together, these issues limit the physiological and mechanistic depth required for publication in a high-impact journal.

Major Concerns

1. Ambiguity in Cell Line Usage

Although the Methods section states the use of both HUES8 and PGP1 cells, the figures, legends, and Results fail to specify which cell line was used in each experiment. This raises concerns about reproducibility and generalizability.

► Authors must clarify which experiments were performed in which cell lines, and replicate key findings (e.g., ATP rescue, PDHB KD) in the second line.

2. Mechanistic Inference of ATP \rightarrow BAF is Not Fully Supported

Figure 5A shows that ATP rescues DE differentiation in PDHB \pm or BrPA-treated cells, and that this rescue is blocked by BAF inhibitors. While this is consistent with the proposed model, it does not exclude other ATP-dependent regulators of chromatin or differentiation.

► Authors should directly assess BAF complex activity in response to ATP changes (e.g., BRG1 occupancy at DE loci, ATPase activity, or nucleosome repositioning assays).

► Alternatively, loss-of-function (e.g., BRG1 KO) combined with ATP rescue would strengthen causal inference.

3. Lack of Direct Metabolic Flux Measurements

The metabolic switch is deduced from static metabolite levels and inhibitor use. These are suggestive but not conclusive.

► Authors should include stable isotope tracing (e.g., ^{13}C -glucose or ^{13}C -pyruvate) to directly confirm increased flux through the TCA cycle during differentiation and after TGF β /PDHB activation.

4. No Exploration of In Vivo or Cross-Lineage Relevance

All findings are based on in vitro differentiation of human iPSCs to endoderm. It remains unknown whether this TGF β –PDHB–ATP–BRG1 axis operates in in vivo contexts, or other lineages (mesoderm, ectoderm), where PDHB expression and mitochondrial flux are also relevant.

► Analysis of publicly available embryo scRNA-seq datasets could help address in vivo relevance.

► Directed differentiation to mesoderm or ectoderm in PDHB \pm or BrPA-treated cells would assess lineage specificity.

5. Statistical Concerns

The manuscript reports the use of t-tests throughout, including in experiments with three or more groups, where ANOVA with appropriate post-hoc correction would be more statistically appropriate.

► Authors should re-analyze all multi-group comparisons using ANOVA or other appropriate statistical frameworks.

Minor Concerns

1. Figure legends should specify which cell line was used in each experiment.

2. The Western blot data are not accompanied by quantification or normalization to loading controls.

3. Clarify terminology: In some sections, PDH and PDHB are used interchangeably. Clearly distinguish PDHB as a subunit and gene target.

4. Figures 5–6 could benefit from improved resolution and annotations.

Version 1:

Reviewer comments:

Reviewer #1

(Remarks to the Author)

All previous concerns have been addressed.

Reviewer #2

(Remarks to the Author)

The authors have addressed all my questions, and the manuscript is qualified for the publication.

Reviewer #3

(Remarks to the Author)

You have done a great job and attempting to address all of my outlined concerns. I have no further comments. Congrats on a wonderful study.

Response to reviewers' comments

We sincerely thank the reviewers for their insightful comments and constructive suggestions, which have greatly helped us improve the quality of our manuscript. In response to the reviewers' concerns, we have conducted additional experiments, performed further data analyses, and revised the manuscript to clarify the statements in the previous version. The new data and results mainly include: (1) establishing BRG1-inducible degradation and ATPase activity mutant systems to investigate the requirement of its ATPase activity in ATP/BRG1-mediated DE differentiation; (2) generating a stable PDHB-overexpressing cell line and performing ¹³C-glucose tracing to confirm enhanced mitochondrial pyruvate utilization; (3) reproducing key findings in another hPSC line (PGP1); (4) performing ChIP-qPCR to assess BRG1 occupancy changes at DE-associated loci; (5) analyzing single-cell RNA-seq data from human pre-gastrulation embryos to support the *in vivo* relevance of our model.

We hope that our revisions can address the concerns satisfactorily. With regard to the specific comments, our point-by-point responses are listed below:

Reviewer #1:

This manuscript by Meng et al. provides compelling evidence that glucose metabolism, specifically, mitochondrial pyruvate flux governed by TGF β -induced PDHB expression, plays a pivotal role in definitive endoderm (DE) differentiation from human pluripotent stem cells (hPSCs). The authors demonstrate that ATP generated through this metabolic reprogramming enhances the activity of the BRG1/BAF chromatin remodeling complex, thereby promoting chromatin accessibility at DE-related gene loci. Through a comprehensive combination of CRISPR screening, small molecule perturbations, metabolite supplementation, RNA-seq, ATAC-seq, and ChIP-seq, the study establishes a mechanistic link between extracellular signaling, cellular metabolism, epigenetic remodeling, and cell fate determination. The study employs innovative approaches to uncover a previously underappreciated role of TGF β -PDHB signaling in directing glucose metabolism toward mitochondrial pyruvate oxidation. However, several concerns should be addressed prior to publication:

Response: We sincerely appreciate the reviewer's positive and encouraging comments, particularly the recognition that "*The study employs innovative approaches to uncover a previously underappreciated role of TGF β -PDHB signaling in directing glucose metabolism toward mitochondrial pyruvate oxidation*". We have accordingly performed additional experiments and analyses and believe the revised version should be significantly strengthened.

1. While the study implicates BRG1 as a downstream effector of ATP-mediated chromatin remodeling, direct evidence of its ATPase activity modulation is lacking. Incorporating experiments with a dominant-negative or ATPase-dead BRG1 mutant would strengthen the causal relationship.

Response: We thank the reviewer for this insightful suggestion. We agree that direct perturbation of BRG1's ATPase activity is critical to establish the causal link between ATP levels and BRG1 function. Given that BRG1 knockdown disrupts pluripotency maintenance in human^{1, 2}, we employed the dTAG-induced degradation system for subsequent experiments³. First, we constructed dTAG-inducible BRG1 degradation ESC lines to circumvent the potential effects of constitutive BRG1 loss on pluripotency maintenance. Then we rescued with wild-type or ATPase-dead (K798R^{4, 5}) BRG1 to investigate whether the ATP influenced DE differentiation by ATPase domain of BRG1.

Western blotting confirmed complete BRG1 degradation within 8 hours of dTAG induction in these two HUES8-derived clones (Fig. 5E). Consistent with the BRM/BRG1 inhibitor experiments, BRG1 loss during DE differentiation impaired endoderm formation (Fig. 5F). Bright-field imaging further revealed that, while control cells lost their colony morphology during differentiation, dTAG-treated BRG1-depleted cells retained compact and colony-like morphology, indicating that they had not exited the pluripotent state.

Fig. 5E

Fig. 5F

Then we further overexpressed wild-type and ATPase-dead (K798R) BRG1 in dTAG-induced BRG1 knockout hESCs. The results showed that only wild-type BRG1 was able to rescue DE differentiation, whereas the ATPase-dead mutant failed (Fig. 5I). This result indicates that BRG1 regulates the differentiation process via its ATPase activity.

Fig. 5I

After BRG1 depletion, we supplemented ATP to examine whether it could rescue the DE differentiation defect caused by BRG1 knockout. The results showed that ATP supplementation failed to restore DE differentiation, further supporting that BRG1 is essential for ATP-mediated enhancement of DE differentiation efficiency (Fig. 5H).

Fig. 5H

Additionally, we employed BRM014, a specific inhibitor of the ATPase activity of BRM/BRG1⁶, to confirm the requirement of BRG1's ATPase function in ATP-mediated differentiation. Consistent with previous results, exogenous ATP supplementation failed to promote DE differentiation when BRG1's ATPase activity was inhibited, indicating that ATP exerts its effect on differentiation through the ATPase function of BRG1 (Fig. S8A).

Fig. S8A

Together, these complementary approaches—pharmacological inhibition, inducible degradation, and ATPase-dead mutant analysis—provide strong evidence that BRG1 functions as an ATP downstream effector through its ATPase activity, thereby regulating the DE differentiation process.

2. It remains unclear whether PDHB overexpression or activation is sufficient to enhance DE differentiation. Addressing this would clarify whether PDHB acts as a limiting factor in this process.

Response: We appreciate the reviewer’s constructive suggestion. To determine whether PDHB activation or overexpression is promotive for DE differentiation, we employed two complementary approaches.

First, we established PDHB-overexpressing ESC lines (Fig. 4F) and performed DE differentiation. PDHB overexpression markedly enhanced the conversion of pyruvate to acetyl-CoA, thereby facilitating its entry into the TCA cycle and promoting ATP production (Fig. 4G).

Fig. 4F

Fig. 4G

During DE differentiation, PDHB overexpression significantly increased differentiation efficiency (Fig. 4H). Notably, when the Activin A concentration was reduced from the standard 100 ng/mL to lower levels, PDHB overexpression further enhanced DE differentiation, suggesting that PDHB activity serves as a limiting factor

under reduced signaling conditions (Fig. 4I).

Fig. 4H

Fig. 4I

Next, we also used sodium dichloroacetate (DCA; APExBio, Cat# B7174), a specific inhibitor of pyruvate dehydrogenase kinase (PDK)⁷, which negatively regulates pyruvate dehydrogenase (PDH) activity through phosphorylation^{8, 9}, and acts as a pharmacological activator of PDH activity^{10, 11} to confirm the results of PDHB overexpression. The results showed that DCA treatment elevated acetyl-CoA levels and significantly enhanced DE differentiation efficiency (Fig. S2B-C).

Fig. S2B

Fig. S2C

Together, PDHB overexpression (genetic manipulation) and DCA treatment (pharmacological activation) demonstrate that PDHB overexpression or activation is able to enhance DE differentiation, suggesting that PDHB acts as a limiting factor in this process.

3. Some figures, particularly Fig. 6 depicting ATAC-seq and BRG1 ChIP-seq overlap, are densely annotated. Simplifying these visuals and including schematic models would improve accessibility and reader comprehension.

Response: We thank the reviewer for this valuable suggestion. In response, we have simplified Figure 6 by reorganizing the ATAC-seq and BRG1 ChIP-seq overlap panels and removing redundant annotations to enhance visual clarity.

4. Terminology such as “mainstream glucose metabolism” versus “PPP” should be clearly defined earlier in the manuscript to avoid ambiguity and improve clarity.

Response: We appreciate the reviewer’s thoughtful comment. To improve clarity and ensure terminological consistency, we have revised the manuscript to explicitly define the terms “mainstream glucose metabolism” and “PPP pathway” (in line 165-168). Specifically, “mainstream glucose metabolism” is now defined as the glycolytic flux leading to pyruvate generation and subsequent mitochondrial oxidation via the PDH–TCA axis, whereas “PPP” refers to the parallel branch responsible for NADPH and ribose-5-phosphate production for biosynthetic processes.

Reviewer #2 (Remarks to the Author):

The authors revealed a metabolic switch with decreased lactate production and increased mitochondrial pyruvate metabolism, which is controlled by TGF β -activated PDHB, is necessary for the definitive endoderm differentiation from human pluripotent stem cells. The experiments data, especially the pyruvate rescue and ATP rescue, are convincing and impressive, and the following are the additional points that can be addressed.

Response: We sincerely appreciate the reviewer's comment that "*The experiments data, especially the pyruvate rescue and ATP rescue, are convincing and impressive*". We have accordingly performed additional experiments and analyses and believe the revised version should be significantly strengthened.

In line 55-56 in the introduction, the glucose metabolism is also required for the cell type specification of TE during blastocyst development.

Response: We thank the reviewer for this insightful comment. We have now revised the Introduction to include this important point. In the revised manuscript, we have added the following sentences to the Introduction: "*Moreover, during this process, both the differentiation of the inner cell mass (ICM) into the three germ layers and the specification of the trophectoderm (TE) are highly dependent on glucose metabolism. Notably, glucose deprivation causes a severe developmental arrest and a significant loss of CDX2-positive TE cells, highlighting the essential role of glucose metabolism in early embryogenesis¹².*"

For figure 1, the authors can add one or two sentence to explain the rationale of using CXCR4 as the readout of DE differentiation in their CRISPR screening.

Response: We appreciate this helpful suggestion. In the revised manuscript, we have added that: "*In CRISPR screening, we used CXCR4 expression as a quantitative readout to evaluate DE differentiation efficiency. As a well-established surface marker of definitive endoderm, CXCR4 has been extensively utilized in previous studies to assess DE differentiation efficiency from pluripotent stem cells^{13, 14}.*"

For figure 1, whether glucose metabolism inhibition leads to compensation of other lineage differentiation, such as ectoderm and mesoderm, or it generally inhibits differentiation and pluripotency exit in this 2D differentiation system. It was discussed in the embryoid body section but not here. Also, the toxicity effect data upon BrPA/2DG treatment was mentioned but not shown.

Response: We appreciate the reviewer's valuable suggestion. To evaluate the impact of

glucose metabolism inhibition on the differentiation of other germ layers, we performed embryoid body (EB) differentiation to assess the effects of BrPA treatment across all three germ layers. The results showed that BrPA treatment slightly impaired the expression of mesendodermal and ectodermal markers (Fig. S2B, in revised manuscript). In Figure 1, we specifically focused on the effect of glucose metabolism inhibition on DE differentiation; therefore, data on mesoderm and ectoderm differentiation were presented somewhere else.

In addition, we performed directed differentiation toward ectoderm and mesoderm lineages under BrPA treatment. For ectoderm differentiation¹⁵, ectoderm markers (*NESTIN*, *PAX6*, *SOX1*) were examined on day 3 and 7. Under BrPA treatment, ectodermal markers remained largely unchanged at day 3 but were markedly reduced by day 7 (Fig. S2C). Mesoderm differentiation was induced using a previously established cardiomyocyte differentiation protocol¹⁶. The expression of cardiac-specific markers (*TNNT2*, *GATA4*, *MYH6*) was impaired in BrPA-treated cells, indicating impaired differentiation toward the mesodermal cardiac lineage (Fig. S2D).

Collectively, these results suggest that inhibition of glucose metabolism generally suppresses both ectodermal and mesodermal differentiation.

Fig. S2C

Fig. S2D

For inhibitor usage, we applied concentrations within the non-cytotoxic range. Cliff and colleagues selected 2.2 mM as the final concentration of 2-DG¹⁷. In our study, we used 1 mM and 2 mM as the working concentration for 2-DG. To assess potential cytotoxicity, we measured the apoptosis rate in BrPA or 2-DG-treated, and PDHB^{+/-} cells using the Annexin V-FITC/PI Apoptosis Detection Kit. Our results showed no significant increase in apoptosis compared to the control, indicating that the inhibitors we used did not affect cell survival (Fig. S1D).

Fig. S1D

For figure 2, the pyruvate and glutamine rescue experiments are interesting and convincing in elucidating the role of glucose metabolism in DE differentiation. Can they also rescue the slight changes of ectoderm differentiation in Fig 2I?

Response: We sincerely thank the reviewer for this insightful comment. As shown in Fig. S2B (in revised manuscript), inhibition of glucose metabolism by BrPA treatment slightly suppressed ectoderm differentiation. Following the reviewer's suggestion, we performed rescue experiments by supplementing exogenous pyruvate, glutamine, and ATP during embryoid body (EB) formation. Consistent with the results observed in DE differentiation, supplementation with these metabolites effectively restored ectoderm

differentiation efficiency impaired by glucose metabolic inhibition (Fig. S2E).

Fig. S2E

In line 276-279, the authors should better explain the difference of mesoendoderm and definitive endoderm to make it coherent through the article.

Response: We appreciate the reviewer's suggestion. During pluripotent stem cell differentiation, cells can directly commit to the ectodermal lineage. However, for mesoderm and endoderm lineages, cells first transition through a mesendodermal stage, marked by the expression of *EOMES*, *MIXL1*, and *T* (Brachyury), before further differentiating into either mesoderm or definitive endoderm (DE)¹⁸.

In Figure 5E (in revised manuscript), we observed that PDHB depletion (PDHB^{+/-}) resulted in an increase in mesendodermal markers. To further investigate this process, we analyzed gene expression dynamics at three time points (Day 3, 5, and 9) during embryoid body (EB) differentiation. PDHB^{+/-} cells exhibited a delayed downregulation of pluripotency genes and a postponed peak in mesendodermal gene expression: mesendoderm markers peaked around Day 5 in wild-type cells, the PDHB^{+/-} cells reached this stage around Day 9 or later (Fig. S5G). This suggests that PDHB^{+/-} impairs both the exit from pluripotency and the initial mesendodermal differentiation. Notably, the upregulation of DE markers was also significantly delayed in PDHB^{+/-} cells. In revised manuscript, we also provided a further explanation for this phenotype.

Can the authors use a PDH inhibitor to show ATP decrease and reduced DE differentiation, which can be rescued by ATP, in addition to the general glycolysis

inhibitor?

Response: We thank the reviewer for this excellent suggestion. To specifically inhibit PDH activity, we used CPI-613 (Selleck S2776), a mitochondrial enzyme inhibitor that blocks pyruvate dehydrogenase activity¹⁹. Similar to the results observed in PDHB^{+/-} cells, CPI-613 treatment led to a dose-dependent reduction in DE differentiation efficiency, accompanied by a significant decrease in intracellular ATP levels. Importantly, both ATP content and differentiation efficiency were successfully rescued by exogenous ATP supplementation (Fig. S7C), further confirming that PDH-regulated metabolic changes influence DE differentiation via ATP availability.

Fig. S7C

For fig 5, the ATP rescue experiments are impressive. Can supplemented ATP be permeable to the cell membrane to change the intracellular ATP level, and is there other reports or this is the first time to show ATP can participate in the ATP-dependent chromatin remodeling processes. These points can be at least discussed in the discussion section.

Response: We sincerely thank the reviewer for this constructive comment. ATP is a charged and hydrophilic molecule that is difficult to freely cross the plasma membrane. Moreover, no plasma membrane-associated ATP transporter has been identified. Therefore, whether exogenously supplied ATP can enter the cell and regulate intracellular energy levels is a critical question.

Several earlier studies have demonstrated that ATP can be both released from and taken up by cells^{20, 21, 22}. However, the exact mechanisms underlying this process remained unclear due to technical limitations. Recent work using nonhydrolyzable fluorescent ATP (NHF-ATP) tracing methods has shown that multiple cell types, including human lung cancer cell A549 and human non-small cell lung cancer (NSCLC) cells, can internalize extracellular ATP via macropinocytosis, leading to an increase in intracellular ATP levels^{23, 24, 25, 26}. These findings provide strong evidence for the internalization of extracellular ATP and suggest that cellular energy status can be modulated through exogenous ATP supplementation.

In our experiments, we directly measured intracellular ATP levels and confirmed that ATP supplementation successfully restored ATP concentrations in DE cells (Fig. S8A). This restoration occurred both under normal conditions and when BRG1 ATPase activity was inhibited, further supporting the idea that extracellular ATP can influence intracellular ATP levels and, consequently, impact cellular processes dependent on ATP availability.

Regarding the role of ATP in ATP-dependent chromatin remodeling, we acknowledge that our study is not the first to report such an effect. Previous structural and biochemical studies of the BAF complex have demonstrated that ATP binding and hydrolysis at the ATPase domain are essential for chromatin remodeling activity. This has been validated through ATPase activity assays and restriction enzyme accessibility assays (REAA) (shown below)²⁷. Together, these findings provide mechanistic support for our results, which show that ATP-mediated activation of BRG1 facilitates chromatin accessibility and contributes to cell differentiation.

[FIGURE REDACTED]

Some of the grammar should be polished such as:

Line 139 To further validate the promotional role

Line 144 evident by the decreased acetyl-CoA

Line 169 Pyruvate entering TCA cycle facilitates DE differentiation

TCA cycle should be the TCA cycle.

Response: We appreciate the reviewer's careful review. In response, we have made the following revisions to improve clarity and readability:

Line 139: "To further validate the promotional role" has been revised to "To further validate the promotive role" for clarity.

Line 144: "Evident by the decreased acetyl-CoA" has been revised to "Evident from the decreased acetyl-CoA" to improve the phrasing.

Line 169: "Pyruvate entering TCA cycle facilitates DE differentiation" has been revised to "Pyruvate entering the TCA cycle facilitates DE differentiation" to ensure correct article usage.

Reviewer #3 (Remarks to the Author):

Summary

Meng et al. investigates how glucose metabolism regulates endoderm differentiation from human iPS cells. The authors demonstrate that mitochondrial pyruvate metabolism is required for ATP production, which in turn supports the chromatin remodeling activity of the BRG1-containing BAF complex. They further link this metabolic switch to TGF β signaling through transcriptional regulation of PDHB by SMAD2/3. Loss of PDHB impairs DE differentiation, which can be partially rescued by exogenous ATP, and this rescue is blocked by BAF complex inhibition. These findings are supported by extensive molecular analyses including CRISPR screens, metabolite measurements, ATAC-seq, ChIP-seq, and RNA-seq.

While the study makes a compelling case for the involvement of mitochondrial metabolism in endoderm fate commitment, several major conceptual, technical, and interpretational limitations lessen its current suitability for publication in Nature Communications.

Conceptual Impact and Novelty / Overall Big Picture Issues

The manuscript brings forward a conceptually appealing model that connects growth factor signaling (TGF β), metabolic state (pyruvate flux \rightarrow ATP), and chromatin remodeling (BAF complex) to orchestrate cell fate decisions in human-iPS cells. The novelty resides in linking ATP-dependent chromatin remodeling to a specific metabolic switch regulated by PDHB, as a direct target of SMAD2/3. This integration of metabolism and epigenetics during germ layer specification is timely and of interest to a broad audience.

However, several gaps undermine the model's completeness and generalizability:

- The central mechanism (ATP \rightarrow BAF) is not fully validated, and alternative ATP-dependent processes are not addressed.
- The metabolic reprogramming is inferred, not directly measured via flux analysis.
- Findings are derived from a single cell line, limiting generalizability.
- The in vivo relevance is not addressed at all.

Together, these issues limit the physiological and mechanistic depth required for publication in a high-impact journal.

Response: We sincerely appreciate the reviewer's evaluation of our work, particularly the recognition that our study "*brings forward a conceptually appealing model*" and highlights the novelty of linking TGF β signaling, mitochondrial metabolism, and

chromatin remodeling in endoderm fate commitment. In response to the reviewer's insightful comments, we have performed additional experiments and analyses to address the raised concerns, and we believe that these revisions have substantially strengthened the mechanistic and conceptual depth of the study.

Major Concerns

1. Ambiguity in Cell Line Usage

Although the Methods section states the use of both HUES8 and PGP1 cells, the figures, legends, and Results fail to specify which cell line was used in each experiment. This raises concerns about reproducibility and generalizability.

➤ Authors must clarify which experiments were performed in which cell lines, and replicate key findings (e.g., ATP rescue, PDHB KD) in the second line.

Response: We thank the reviewer for this important comment. In our study, the majority of experiments were performed using the human embryonic stem cell line HUES8, while selected key experiments were independently validated in another cell line PGP1, such as the Fig. 1E (in revised manuscript). We have now clearly indicated the specific cell line used in each figure and corresponding legend in the revised manuscript.

In the revised manuscript, we also replicated more critical experiments in PGP1 cells. Specifically, we used BrPA to inhibit global glycolytic activity and CPI-613 (Selleck S2776) as a selective PDH inhibitor¹⁹. Consistent with the results obtained in HUES8 cells (in answers for reviewer#2, Fig. S7C), both treatments led to decreased intracellular ATP levels and impaired definitive endoderm differentiation, while exogenous ATP supplementation successfully restored ATP levels and differentiation efficiency. Furthermore, inhibition of BRG1 ATPase activity (by BRM/BRG1 inhibitor, BRM014) abolished this ATP-mediated rescue effect (Fig. S8C), supporting that the observed phenomena are conserved across different stem cell lines.

Fig. S8C

2. Mechanistic Inference of ATP → BAF is Not Fully Supported

Figure 5A shows that ATP rescues DE differentiation in PDHB+/- or BrPA-treated cells,

and that this rescue is blocked by BAF inhibitors. While this is consistent with the proposed model, it does not exclude other ATP-dependent regulators of chromatin or differentiation.

➤ Authors should directly assess BAF complex activity in response to ATP changes (e.g., BRG1 occupancy at DE loci, ATPase activity, or nucleosome repositioning assays).

Response: We thank the reviewer for the insightful suggestion. Previous structural and biochemical studies of the BAF complex have demonstrated that ATP binding and hydrolysis at the ATPase domain are essential for chromatin remodeling activity, being validated through ATPase activity assays and restriction enzyme accessibility assays (REAA)²⁷. To investigate whether ATP influences DE differentiation by modulating BRG1 occupancy at DE-associated loci in our experiment, we performed ChIP-qPCR on several key regulators of DE differentiation. The results revealed that PDHB heterozygous deletion led to a marked decrease in BRG1 enrichment at these loci (including *FOXI1*, *TCF4*, *SMAD3* and *SOX17*) (Fig.6I), concomitant with the reduced chromatin accessibility observed in ATAC-seq. These findings provide mechanistic insight into how reduced chromatin accessibility caused by BRG1 dysfunction is linked to impaired DE differentiation, highlighting the role of ATP and PDHB in regulating BRG1 activity at these loci.

Fig. 6I

➤ Alternatively, loss-of-function (e.g., BRG1 KO) combined with ATP rescue would strengthen causal inference.

Response: We thank the reviewer for this insightful suggestion. We agree that direct perturbation of BRG1's ATPase activity is critical to establish the causal link between ATP levels and BRG1 function. Given that BRG1 knockdown disrupts pluripotency maintenance in human, we employed the dTAG-induced degradation system for subsequent experiments^{1, 2}. First, we constructed dTAG-inducible BRG1 degradation cell lines to circumvent the potential effects of constitutive BRG1 loss on pluripotency maintenance. Then we rescued with wild-type or ATPase-dead (K798R^{4, 5}) BRG1 to

investigate whether the ATP influenced DE differentiation by ATPase domain of BRG1.

Western blotting confirmed complete BRG1 degradation within 8 hours of dTAG induction (Fig. 5E). Consistent with the BRM/BRG1 inhibitor experiments, BRG1 loss during DE differentiation impaired endoderm formation (Fig. 5F). Then we further overexpressed wild-type and ATPase-dead (K798R) BRG1 in dTAG-induced BRG1 knockout hESCs. The results showed that only wild-type BRG1 was able to rescue DE differentiation, whereas the ATPase-dead mutant failed (Fig. 5I). These results indicate that BRG1 regulates the differentiation process via its ATPase activity.

After BRG1 deletion, we supplemented ATP to examine whether it could rescue the DE differentiation defect caused by BRG1 knockout. The results showed that ATP supplementation failed to restore DE differentiation, further supporting that the ATPase activity of BRG1 is essential for ATP-mediated enhancement of DE differentiation efficiency (Fig. 5H). Additionally, we employed BRM014, a specific inhibitor of the ATPase activity of BRM/BRG1⁶, to confirm the requirement of BRG1's ATPase function in ATP-mediated differentiation. Consistent with previous results, exogenous ATP supplementation failed to promote DE differentiation when BRG1's ATPase activity was inhibited (Fig. S8A), indicating that ATP exerts its effect on differentiation

through the ATPase function of BRG1.

Together, these complementary approaches—pharmacological inhibition, inducible degradation, and ATPase-dead mutant analysis—provide strong evidence that BRG1 functions as an ATP downstream effector through its ATPase activity, thereby regulating the DE differentiation process.

3. Lack of Direct Metabolic Flux Measurements

The metabolic switch is deduced from static metabolite levels and inhibitor use. These are suggestive but not conclusive.

➤ Authors should include stable isotope tracing (e.g., ^{13}C -glucose or ^{13}C -pyruvate) to directly confirm increased flux through the TCA cycle during differentiation and after TGF β /PDHB activation.

Response: We thank the reviewer for the good advice. In response, we performed [U- ^{13}C]-glucose tracing experiments to analyze metabolism alterations during differentiation and PDHB activation. During DE differentiation, we indeed observed a decrease in the ^{13}C labeling ratio of several glycolytic intermediates, including fructose-1,6-bisphosphate (FBP), 1,3-bisphosphoglycerate (1,3-BPG), 3-phosphoglycerate / 2-phosphoglycerate (3PG/2PG), phosphoenolpyruvate (PEP), as well as the end product lactate. These results further confirm the reduced glycolytic activity following DE differentiation.

Next, we established a PDHB-overexpressing hESC line (Fig. 4F) to examine changes in TCA cycle intermediates upon PDHB overexpression. In terms of metabolic changes, PDHB overexpression increased glucose-derived labeling in key TCA cycle metabolites, including citrate, aconitate, succinate, and succinyl-CoA (Fig. S6A-B). Collectively, our ¹³C metabolic flux analysis demonstrates that PDHB overexpression promotes a metabolic shift toward elevated TCA cycle activity, providing a mechanistic explanation for the enhanced DE differentiation.

Fig. 4F

Fig. S6A

Fig. S6B

4. No Exploration of In Vivo or Cross-Lineage Relevance

All findings are based on in vitro differentiation of human iPSCs to endoderm. It remains unknown whether this TGF β -PDHB-ATP-BRG1 axis operates in in vivo contexts, or other lineages (mesoderm, ectoderm), where PDHB expression and mitochondrial flux are also relevant.

➤ Analysis of publicly available embryo scRNA-seq datasets could help address in vivo relevance.

Response: We thank the reviewer for this constructive suggestion. To further address this issue, we analyzed the single-cell RNA-seq dataset of 3D-cultured human pre-gastrulation embryos²⁸, which includes multiple cell types at the gastrulation stage, such as the embryonic disc, amnion, basement membrane, primary and primate-specific secondary yolk sac, anterior-posterior polarity formation, and primitive streak anlage (PSA)²⁹. Based on these data, we examined the expression patterns of genes involved in glycolysis, the TCA cycle, oxidative phosphorylation, and the BAF chromatin remodeling complex across different embryonic tissues.

Our analysis revealed that during the developmental transition from the inner cell mass (ICM) to epiblast (EPI) and subsequently to PSA-EPI, glycolytic gene expression—including canonical glycolytic genes such as *GAPDH*, *GPI*, *HK2*, and *PDHB*—was markedly decreased, whereas genes related to the TCA cycle and oxidative phosphorylation were significantly upregulated. These results indicate a clear

metabolic shift toward mitochondrial oxidative metabolism during this stage. Furthermore, genes associated with the BAF complex, such as *SMARCA4 (BRG1)*, *BRD7*, *BRD9*, and *ARID1A*, also exhibited a moderate increase in expression throughout the ICM-EPI-PSA-EPI differentiation process. Together, these findings suggest that the metabolic and epigenetic changes described in our study also occur during normal human embryonic development. Notably, such a metabolic transition was not observed in lineages differentiating toward extraembryonic fates. (CTBs: cytotrophoblasts; STBs: syncytiotrophoblasts; EVTs: extravillous cytotrophoblasts; PrE: primitive endoderm/hypoblast)

Jiang L et al., Stem Cell Reports 2024 Fig. S9A

[FIGURE REDACTED]

► Directed differentiation to mesoderm or ectoderm in PDHB^{+/-} or BrPA-treated cells would assess lineage specificity.

Response: We appreciate the reviewer's valuable suggestion. To assess lineage specificity, we performed ectoderm differentiation using the protocol described in our previous publication¹⁵. On day 3 and 7 of differentiation, we examined the expression of pluripotency genes and ectoderm-associated markers, including *NESTIN*, *PAX6*, and *SOX1*. Under BrPA treatment, ectodermal gene expression showed no significant reduction at day 3 but was markedly decreased at day 7. Consistently, the PDHB^{+/-} cell line exhibited suppression of ectodermal gene expression at both time points. Regarding pluripotency genes, we observed a slightly increase at day 3, reflecting delayed differentiation. However, this increase diminished at day 7, indicating that despite impaired ectoderm differentiation, the cells still exited the pluripotent state.

To further clarify the effect on mesoderm differentiation, we performed directed cardiomyocyte differentiation¹⁶. We observed that the expression of cardiac-specific markers (*TNNT2*, *GATA4*, *MYH6*, *MEF2C*) was impaired in BrPA treated and PDHB^{+/-} cell line, indicating an impaired differentiation phenotype toward the mesodermal cardiac lineage.

5. Statistical Concerns

The manuscript reports the use of t-tests throughout, including in experiments with three or more groups, where ANOVA with appropriate post-hoc correction would be more statistically appropriate.

► Authors should re-analyze all multi-group comparisons using ANOVA or other appropriate statistical frameworks.

Response: We appreciate the reviewer's insightful comment regarding the statistical analysis. In response, we have re-analyzed all multi-group comparisons. For figures with three or more groups, statistics were calculated using one-way ANOVA, followed by Dunnett's multiple comparisons test. Two-way ANOVA was used when two variables were involved, followed by Tukey's multiple comparisons test. For comparisons between two groups, paired t-tests were applied directly. The revised statistical analyses and updated results are now included in the manuscript and corresponding figure legends.

Minor Concerns

1. Figure legends should specify which cell line was used in each experiment.

Response: We thank the reviewer for pointing this out. In the revised manuscript, we have updated the figure legends to clearly specify which cell line was used in each experiment, ensuring better clarity for readers.

2. The Western blot data are not accompanied by quantification or normalization to

loading controls.

Response: We appreciate the reviewer's comment regarding the Western blot data. In response, we have now included quantification of the Western blot signals and have normalized the data to appropriate loading controls (e.g., α -tubulin or H3) for all relevant experiments. These updates are described in the figure legends.

3. Clarify terminology: In some sections, PDH and PDHB are used interchangeably. Clearly distinguish PDHB as a subunit and gene target.

Response: We thank the reviewer for the careful review. We have revised the manuscript to clarify the terminology. Specifically, we now distinguish PDHB as the subunit of pyruvate dehydrogenase (PDH) and the gene encoding this subunit. The revised manuscript clearly differentiates these terms to avoid any confusion.

4. Figures 5–6 could benefit from improved resolution and annotations.

Response: We appreciate the reviewer's suggestion to improve the resolution and annotations of Figures 5 and 6. In response, we have enhanced the resolution of these figures and added additional annotations to improve clarity and accessibility.

Reference

1. Ho L, *et al.* An embryonic stem cell chromatin remodeling complex, esBAF, is essential for embryonic stem cell self-renewal and pluripotency. *Proc Natl Acad Sci U S A* **106**, 5181-5186 (2009).
2. Zhang X, *et al.* Transcriptional repression by the BRG1-SWI/SNF complex affects the pluripotency of human embryonic stem cells. *Stem Cell Reports* **3**, 460-474 (2014).
3. Nabet B, *et al.* The dTAG system for immediate and target-specific protein degradation. *Nat Chem Biol* **14**, 431-441 (2018).
4. Khavari PA, Peterson CL, Tamkun JW, Mendel DB, Crabtree GR. BRG1 contains a conserved domain of the SWI2/SNF2 family necessary for normal mitotic growth and transcription. *Nature* **366**, 170-174 (1993).
5. Dykhuizen EC, *et al.* BAF complexes facilitate decatenation of DNA by topoisomerase IIalpha. *Nature* **497**, 624-627 (2013).
6. Papillon JPN, *et al.* Discovery of Orally Active Inhibitors of Brahma Homolog (BRM)/SMARCA2 ATPase Activity for the Treatment of Brahma Related Gene 1 (BRG1)/SMARCA4-Mutant Cancers. *J Med Chem* **61**, 10155-10172 (2018).
7. Tso SC, *et al.* Structure-guided development of specific pyruvate dehydrogenase kinase

inhibitors targeting the ATP-binding pocket. *J Biol Chem* **289**, 4432-4443 (2014).

8. Anwar S, Shamsi A, Mohammad T, Islam A, Hassan MI. Targeting pyruvate dehydrogenase kinase signaling in the development of effective cancer therapy. *Biochim Biophys Acta Rev Cancer* **1876**, 188568 (2021).
9. Whitehouse S, Cooper RH, Randle PJ. Mechanism of activation of pyruvate dehydrogenase by dichloroacetate and other halogenated carboxylic acids. *Biochem J* **141**, 761-774 (1974).
10. Huang J, *et al.* Histone lactylation drives liver cancer metastasis by facilitating NSF1-mediated ferroptosis resistance after microwave ablation. *Redox Biol* **81**, 103553 (2025).
11. Gao J, *et al.* Dynamic investigation of hypoxia-induced L-lactylation. *Proc Natl Acad Sci U S A* **122**, e2404899122 (2025).
12. Moussaieff A, *et al.* Glycolysis-mediated changes in acetyl-CoA and histone acetylation control the early differentiation of embryonic stem cells. *Cell Metab* **21**, 392-402 (2015).
13. D'Amour KA, Agulnick AD, Eliazar S, Kelly OG, Kroon E, Baetge EE. Efficient differentiation of human embryonic stem cells to definitive endoderm. *Nat Biotechnol* **23**, 1534-1541 (2005).
14. Yi Y, *et al.* Fatty acid synthesis and oxidation regulate human endoderm differentiation by mediating SMAD3 nuclear localization via acetylation. *Dev Cell* **58**, 1670-1687 e1674 (2023).
15. Meng Y, *et al.* Depletion of Demethylase KDM6 Enhances Early Neuroectoderm Commitment of Human PSCs. *Front Cell Dev Biol* **9**, 702462 (2021).
16. Cai L, *et al.* A Human Engineered Heart Tissue-Derived Lipotoxic Diabetic Cardiomyopathy Model Revealed Early Benefits of Empagliflozin. *Adv Sci (Weinh)* **12**, e03173 (2025).
17. Cliff TS, *et al.* MYC Controls Human Pluripotent Stem Cell Fate Decisions through Regulation of Metabolic Flux. *Cell Stem Cell* **21**, 502-516 e509 (2017).
18. Thowfeequ S, Srinivas S. Embryonic and extraembryonic tissues during mammalian development: shifting boundaries in time and space. *Philos Trans R Soc Lond B Biol Sci* **377**, 20210255 (2022).
19. Zachar Z, *et al.* Non-redox-active lipoate derivatives disrupt cancer cell mitochondrial metabolism and are potent anticancer agents in vivo. *J Mol Med (Berl)* **89**, 1137-1148 (2011).

20. Chaudry IH, Gould MK. Evidence for the uptake of ATP by rat soleus muscle in vitro. *Biochim Biophys Acta* **196**, 320-326 (1970).
21. Chaudry IH, Baue AE. Further evidence for ATP uptake by rat tissues. *Biochim Biophys Acta* **628**, 336-342 (1980).
22. Chaudry IH. Does ATP cross the cell plasma membrane. *Yale J Biol Med* **55**, 1-10 (1982).
23. Zhou Y, *et al.* Intracellular ATP levels are a pivotal determinant of chemoresistance in colon cancer cells. *Cancer Res* **72**, 304-314 (2012).
24. Wang X, *et al.* Extracellular ATP, as an energy and phosphorylating molecule, induces different types of drug resistances in cancer cells through ATP internalization and intracellular ATP level increase. *Oncotarget* **8**, 87860-87877 (2017).
25. Qian Y, *et al.* Extracellular ATP is internalized by macropinocytosis and induces intracellular ATP increase and drug resistance in cancer cells. *Cancer Lett* **351**, 242-251 (2014).
26. Qian Y, Wang X, Li Y, Cao Y, Chen X. Extracellular ATP a New Player in Cancer Metabolism: NSCLC Cells Internalize ATP In Vitro and In Vivo Using Multiple Endocytic Mechanisms. *Mol Cancer Res* **14**, 1087-1096 (2016).
27. Mashtalir N, *et al.* A Structural Model of the Endogenous Human BAF Complex Informs Disease Mechanisms. *Cell* **183**, 802-817 e824 (2020).
28. Xiang L, *et al.* A developmental landscape of 3D-cultured human pre-gastrulation embryos. *Nature* **577**, 537-542 (2020).
29. Jiang L, *et al.* Cell size regulates human endoderm specification through actomyosin-dependent AMOT-YAP signaling. *Stem Cell Reports* **19**, 1137-1155 (2024).